# FS-DFM: Fast and Accurate Long Text Generation with Few-Step Diffusion Language Models

Amin Karimi Monsefi[1,2]*    Nikhil Bhendawade[2]    Manuel R. Ciosici[2]    Dominic Culver[2]

Yizhe Zhang[2]    Irina Belousova[2]

[1]The Ohio State University    [2]Apple

## Abstract

Autoregressive language models (ARMs) deliver strong likelihoods, but are inherently serial: they generate one token per forward pass, which limits throughput and inflates latency for long sequences. Diffusion Language Models (DLMs) parallelize across positions and thus appear promising for language generation, yet standard discrete diffusion typically needs hundreds to thousands of model evaluations to reach high quality, trading serial depth for iterative breadth. We introduce **FS-DFM**, Few-Step Discrete Flow-Matching. A discrete flow-matching model designed for speed without sacrificing quality. The core idea is simple: make the number of sampling steps an explicit parameter and train the model to be consistent across step budgets, so one big move lands where many small moves would. We pair this with a reliable update rule that moves probability in the right direction without overshooting, and with strong teacher guidance distilled from long-run trajectories. Together, these choices make few-step sampling stable, accurate, and easy to control. On language modeling benchmarks, FS-DFM with 8 sampling steps achieves perplexity parity with a 1 024-step discrete-flow baseline for generating 1 024 tokens using a similar-size model, delivering up to 128× faster sampling and corresponding latency/throughput gains. We plan to release code and model checkpoints to facilitate reproducibility and further research. **Code & pretrained checkpoints:** github.com/apple/ml-fs-dfm.

## 1 Introduction

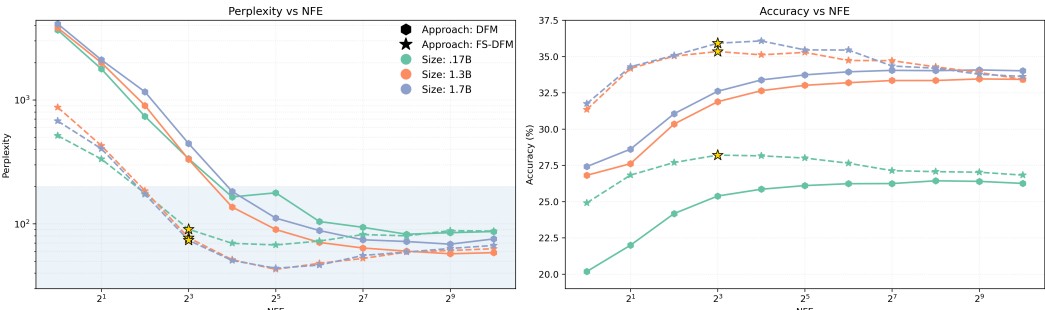

Figure 1: **Generation quality across model sizes** (perplexity and accuracy vs. NFE). FS-DFM reaches the strong-quality regime in few steps across all sizes, while DFM needs far more evaluations. Gold stars (NFE=8) highlight FS-DFM in a few-step regime, with accuracy quickly saturating and entropy converging to similar ranges as steps increase. The average value of entropy for all the models is 7.41 to 8.07.

Autoregressive language models (ARMs) generate sequences by predicting the next token conditioned on the observed prefix, and have achieved remarkable success (Yang et al., 2025; Team et al., 2023; Grattafiori et al., 2024; Li et al., 2025a). In contrast, Diffusion Language Models (DLMs) synthesize text through iterative refinement, providing stable likelihood-grounded training objectives, inherent parallelism across positions, and enhanced controllability through access to global

---

*Work done during the internship at Apple.

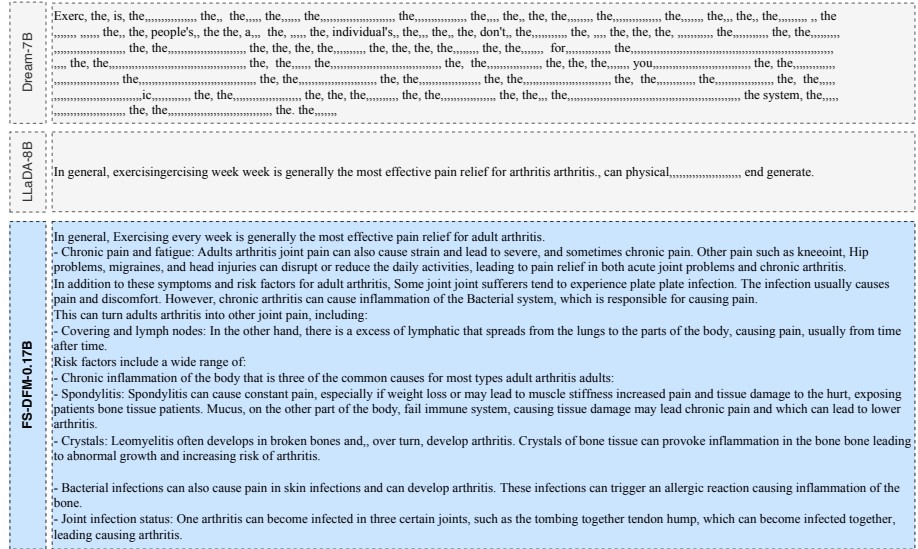

Figure 2: **Eight-step long-horizon generation:** 1 024-token unconditional generation in *8 sampling steps*. FS-DFM (0.17B) successfully produces 1 024 tokens under the 8-step constraint. Despite having 40x more parameters, LLaDA-8B-Instruct and Dream-7B-Instruct's 8-steps generations exhibit trailing blanks and punctuation artifacts (e.g., repeated commas). Generations are truncated. Complete output in Appendix E.5.

context, capabilities that help mitigate exposure bias and naturally accommodate structure beyond left-to-right generation (Nie et al., 2025b; Austin et al., 2021). Despite these complementary advantages, both paradigms face fundamental bottlenecks. ARMs are constrained by sequential decoding, requiring one forward pass per token, which limits throughput and enforces a unidirectional dependency that hinders tasks such as reversal or order-invariant reasoning (Zhao et al., 2025; Schulman et al., 2022; Gong et al., 2025b). Although DLMs parallelize over positions, they typically require tens to hundreds of refinement steps to achieve competitive quality, effectively replacing the autoregressive depth of a single long pass with a stack of iterative model evaluations (Chen & Zhou, 2023; Arriola et al., 2025; Nie et al., 2025a). For example, to march the ARM generation quality, LLaDA requires roughly one inference step per output token (Nie et al., 2025b).

Continuous-space flow-matching models have emerged as a streamlined alternative to conventional score-based diffusion trained around reverse Stochastic Differential Equations (SDEs). Instead of modeling a stochastic reverse process, flow-matching directly trains a continuous normalizing flow via vector-field regression, enabling deterministic probability-flow Ordinary Differential Equation (ODE) sampling. This approach often achieves comparable or superior likelihoods and sample quality with fewer function evaluations and more stable optimization (e.g., through optimal transport or straight-line paths) than SDE-based pipelines (Lipman et al., 2023; Song et al., 2021; Li et al., 2025b; Gong et al., 2023). Discrete Flow-Matching (DFM) extends these advantages to discrete domains by adapting flow-matching to tokenized data, providing flexible probability paths and strong perplexity at scale. However, DFM still requires a substantial number of evaluations along each trajectory, hindering applicability (Gat et al., 2024).

We introduce Few-Step Discrete Flow-Matching (FS-DFM), a new diffusion-based language model built upon the DFM paradigm, which preserves the key advantages of prior diffusion and flow models (bidirectional context utilization and flexible generation) while explicitly mitigating speed and sampling overhead weaknesses. FS-DFM uses DFM's discrete-flow perspective to substantially reduce the number of refinement steps required for high-quality generation. In doing so, FS-DFM narrows the inference efficiency and modeling performance gap between DLMs and ARMs. FS-DFM can reach the quality regime of a 1 024-step DFM with just 8 NFEs (Number of Function Evaluations), 128× fewer steps for comparable quality (Figure 1).

FS-DFM builds on two pillars. First, we enrich the model with the shortcut principle (Frans et al., 2025) together with a self-consistency constraint: the network is conditioned not only on masked tokens but also on the desired step size, and trained so that a single large step agrees with the

composition of multiple smaller steps. We instantiate the shortcut teacher using a Runge–Kutta formulation —we evaluate RK-2 (Heun) and classical RK-4 as ODE estimators— and adopt the one that offers the best accuracy–stability trade-off for large single/few-step updates. Second, a DFM reformulation that models sequences as a Continuous-Time Discrete Markov Chain (CTMC, Campbell et al. (2022)) and a probability velocity is learned to transport a source distribution to the data distribution along a probability path (Gat et al., 2024; Campbell et al., 2024). In this framework, a velocity field advances samples "locally" from $p_t$ to $p_{t+h}$. Instead, FS-DFM learns a *Cumulative Scalar* parameterized by the step size $h$ to enable large and reliable jumps along the probability path. The step-budgeted perspective retains the core benefits of flow-matching: stable training, parallel refinement, and bidirectional context, while elevating few-step generation to a first-class design objective.

To our knowledge, FS-DFM is the first few-step discrete flow–matching approach aimed at long-horizon language modeling. Prior discrete diffusion/flow methods require hundreds or even thousands of refinement steps, often growing with sequence length (Gat et al., 2024; Nie et al., 2025b). FS-DFM works directly in token space with step-aware DFM and a closed-form cumulative scalar update, matching the perplexity of a 1 024-step discrete-flow baseline for 1 024-token generation in just 8 steps—up to **128× faster while preserving long-horizon quality.**

## 2 RELATED WORK

Diffusion models have demonstrated remarkable effectiveness in continuous domains such as images, achieving state-of-the-art quality across diverse tasks (Karimi Monsefi et al., 2025; Zhang et al., 2023; Navard et al., 2024; Rombach et al., 2022). Extending diffusion-style generative modeling to text, Diffusion Language Models (DLMs), have shown impressive task performance coupled with fast inference through parallel generation (Nie et al., 2025b; Ye et al., 2025; Labs et al., 2025). DLMs can be broadly grouped into *continuous* models, which denoise in embedding or latent space, and *discrete* models, which operate directly over tokens. Both categories exhibit the key advantages of diffusion, including bidirectional context, controllability, and parallel refinement (Li et al., 2025b).

Continuous DLMs (Li et al., 2022; Lin et al., 2023; Lovelace et al., 2023) cast text generation as iterative denoising in a continuous embedding or latent space. A forward corruption process perturbs token embeddings or hidden states, while a parameterized denoiser reconstructs clean representations conditioned on the context. Continuous DLMs take advantage of benefits that arise naturally in continuous representations: flexible conditioning and guidance, bidirectional contextualization during refinement, and compatibility with large-scale pretraining or task-specific fine-tuning. Moreover, continuous DLMs benefits from research in diffusion models for other modalities such as images or audio as methods easily port between domains.

Discrete DLMs, on the other hand, model token dynamics either through discrete-time noise schedules or continuous-time Markov formulations. DLMs such as LLaDA (Nie et al., 2025b) and Dream (Ye et al., 2025) match autoregressive models in task performance if allowed sufficient refinement steps. Effectively, they tie step count to sequence length and generation quality (Austin et al., 2021; Gong et al., 2025a; Zheng et al., 2024; Shi et al., 2024; Ou et al., 2025). One line of research accepts this high step count requirement and focuses on speeding up inference through approaches such as confidence-aware parallel decoding, tweaks to re-enable the KV Cache, and restricted attention (Wu et al., 2025; Arriola et al., 2025; Chen et al., 2025b). Another line of research focuses on reducing the number of inference steps through one-/few-step approaches (Gat et al., 2024). Our work falls within this second line of research.

Prior work on one-step language generation primarily focused on continuous-space diffusion models, operating over embeddings rather than discrete tokens. DLM-One (Chen et al., 2025a) demonstrated large speedups on short sequence tasks but did not address long sequence modeling. Similarly, FlowSeq (Hu et al., 2024) operates on continuous embeddings, bypassing token-level likelihoods. Within the discrete family, recent methods typically attain quality with thousands of refinement steps, often scaling with sequence length (Gat et al., 2024; Zhao et al., 2025). SDTT (Deschenaux & Gulcehre, 2025) distills discrete DLMs for faster sampling, yet still uses 16–256 steps and focuses on short-form tasks. In contrast, our method (FS-DFM) operates *in the discrete space*

and explicitly targets *the few-step regime* (1-8 steps). FS-DFM can match 1 024-step discrete-flow baselines in just 8 steps while preserving text quality.

## 3 PRELIMINARIES AND BACKGROUND

We provide a brief summary on flow-matching, following Gat et al. (2024). For the notation used and for details on the theory behind discrete flow-matching, readers are referred to Appendix A.

**Setup.** Fix a target distribution $p_1$ of text, thought of as length $L$ sequences of token ids. The starting point of the flow-matching approach to text generation is a probability path $(p_t)_{0 \le t \le 1}$ that interpolates between a source distribution $p_0$ and the target distribution $p_1$. Given this path, we can sample $X_t \sim p_t$ for $t \in [0, 1]$, which leads to a stochastic process $(X_t)_{0 \le t \le 1}$.

Discrete Flow-Matching (DFM) models the stochastic process $(X_t)_{0 \le t \le 1}$ as a Continuous-Time Markov Chain (CTMC). The evolution of this CTMC can be captured by its *infinitesimal generator* $u_t(\cdot, \cdot)$ for $t \in [0, 1]$. For token sequences $x, y$, the value $u_t(x, y)$ captures the rate of change of the probability that the sequence will change from state $x$ to state $y$ and must satisfy the conditions in proposition A.1. We also have the following equality for small $h$ (see Appendix A.1 for notation)

$$\mathbb{P}(X_{t+h} = y | X_t = x) = \delta_x(y) + h u_t(x, y) + o(h). \tag{1}$$

The goal of DFM is to learn the infinitesimal generator $u_t$. DFM makes the further simplifying assumption (to reduce output dimension size) that $u_t(x, y)$ can be factorized as

$$u_t(x, y) = \sum_i \delta_{\overline{x^i}}(\overline{y^i}) u_t^i(y^i, x). \tag{2}$$

Similarly, the token-level $u_t^i$ must satisfy analogous conditions to proposition A.1. This reduces the output dimension from $|V|^L$ to $|V| \cdot L$, making the learning objective feasible. From this, we then have, for small $h$,

$$\mathbb{P}(X_{t+h}^i = y^i | X_t = x) = \delta_{x^i}(y^i) + h u_t^i(y^i, x) + o(h). \tag{3}$$

This reduces DFM to learning the factors $u_t^i$. It also provides a way of simulating the CTMC and generating samples from noise; given a sample $X_t$ we sample $X_{t+h}^i$ via Euler sampling

$$X_{t+h}^i \sim \delta_{X_t^i} + h \cdot u_t^i(\cdot, X_t) + o(h), \quad i = 1, \dots, L$$

Of course, this depends on the choice of probability path $(p_t)_{0 \le t \le 1}$. Following Gat et al. (2024), start with

$$p_t(x) = \sum_{x_0, x_1} p_t(x | x_0, x_1) \pi(x_0, x_1)$$

where $x_0 \sim p_0$, $x_1 \sim p_1$, and $\pi$ is a joint distribution which relates $p_0$ and $p_1$[1], and

$$p_t(x | x_0, x_1) = \prod_i p_t(x^i | x_0, x_1).$$

Then set the conditional probabilities $p_t(x^i | x_0, x_1)$ to be the convex sum:

$$p_t(x^i | x_0, x_1) := (1 - \kappa_t) \delta_{x_0}(x^i) + \kappa_t \delta_{x_1}(x^i), \tag{4}$$

so that the probability path of the $i$th token position is a linear interpolation between the source and the target distributions. Here, $\kappa_t$ is known as a *scheduler* and must be a monotonically increasing differentiable function $\kappa : [0, 1] \to [0, 1]$ where $\kappa_0 = 0$ and $\kappa_1 = 1$. Using the "Marginalization Trick", one can derive the following description of the factorized velocities $u_t^i$

$$u_t^i(x^i, z) = \frac{\dot{\kappa}_t}{1 - \kappa_t} \left( p_{1|t}(x^i | z) - \delta_z(x^i) \right) \tag{5}$$

where $z$ is a token sequence and

$$p_{1|t}(x^i | z) := \sum_{x_0, x_1} \delta_{x_1}(x^i) p_t(x_0, x_1 | z). \tag{6}$$

---

[1]Namely, $\sum_{x_0} \pi(x_0, x_1) = p_1(x_1)$ and $\sum_{x_1} \pi(x_0, x_1) = p_0(x_0)$

In DFM, the model only needs to learn $p_{1|t}(x^i|z)$. For notational convenience, we also set

$$g(t) := \frac{\dot{\kappa}_t}{1 - \kappa_t}. \tag{7}$$

**Learning the Denoiser.** Given the factorized velocities derived above, the remaining task is to learn $p_{1|t}(\cdot|z)$. We parameterize this conditional distribution using a network $\theta$ that outputs logits:

$$p_{1|t}(x^i|z) = \text{softmax}(\theta^i(z, t))$$

where $\theta^i(z, t)$ denotes the logits for the $i$-th token position.

To train this model, we need an appropriate loss function. We use the Bregman divergence for the loss function (Lipman et al., 2024, Equation 7.31). Starting from the velocity formulation in Equation (5), we can construct a loss that encourages the model to correctly predict $p_{1|t}$. Given a sample trajectory where $x_0 \sim p_0$, $x_1 \sim p_1$ are related through $\pi$, and $x_t$ is sampled from $p_t(x|x_0, x_1)$, the per-token loss at position $i$ is:

$$\mathcal{L}_i(x_1, x_t, t) = -g(t) \left[ p_{1|t}(x_t^i|x_t) - \delta_{x_1^i}(x_t^i) + \left(1 - \delta_{x_1^i}(x_t^i)\right) \log p_{1|t}(x_1^i|x_t) \right] \tag{8}$$

This loss encourages the model to assign high probability to the true target token $x_1^i$ when denoising from $x_t$. The scaling factor $g(t)$ naturally arises from the velocity formulation and ensures proper weighting across different time steps.

## 4 METHOD

Our approach comprises two components: *Step-Aware Discrete Flow-Matching* and a *Cumulative Scalar* update. The first component exposes the step budget $h$ as an explicit control signal within the DFM generator $u_t$ and distills from a shortcut teacher model (Frans et al., 2025), allowing the learner to make large, single/few-step moves that approximate the cumulative effect of many small updates. The second is using a cumulative scalar to aid the model in jumping from a source token to a target token with large single/few-step predictions.

### 4.1 STEP-AWARE DISCRETE FLOW-MATCHING

We expose the *step budget* $h$ as an explicit conditioning signal to the DFM generator so the model learns transitions calibrated to the intended number of sampling steps. To expose the step budget to the model during training, we need a way to produce the Markov chain transition probabilities over large step intervals. However, determining the transition probabilities from the generator $u_t$ is intractable. But, because the transition probabilities satisfy the Kolmogorov equations (Equations (15) and (16)) which are ODEs, we can use numerical methods to approximate the transition probabilities. The role of the teacher model is to approximate the evolution of the Markov chain, to be used as the ground truth for training FS-DFM. Many methods exist for approximating the ODE, each with different trade-offs between accuracy and computational efficiency. We experimented with two Runge-Kutta methods: RK-2 (Heun average) and RK-4 and found that RK-4 greatly improved the results (Figure 3), though at the cost of being more computationally expensive.

**RK-4 ODE solver.** The RK-4 algorithm is presented in Algorithm 1. Define Vel to be the function which computes the velocity from the logits. In particular, given logits $\ell$, current state $x_t$, Vel computes the result of Equation (5):

$$\text{Vel}(\text{softmax}(\ell), x_t, t) := g(t) \left( p_{1|t}(\cdot|x_t) - \delta_{x_t}(\cdot) \right).$$

Given the velocity $u$ from Vel, we apply jump sampling (see Appendix A.3) to obtain $x = \text{Jump}(x_t, u, h)$ where $h$ is our desired time step. For a given time $t$ and time step $h$, RK-4 computes the model logits at $t$ and $t + h/2$ using Vel and Jump as defined above. Note that we use Jump only to obtain the relevant state at $t + h/2$ in order to compute the logits there. RK-4 then averages these logits. We include an detailed explanation of RK-2 in Appendix B.2.

**EMA teacher for stability.** Because training is non-stationary, using the *current* student as the teacher causes the shortcut target to drift with parameter updates, which destabilizes large steps

---

**Algorithm 1** Shortcut RK-4

---

**Require:** tokens $x_t$, time $t$, step $h$; Model $\theta$; velocity Vel; CTMC jumper Jump; *use_ema* flag
1: $h' \leftarrow h/2$;  $t_{\mathrm{mid}} \leftarrow t + h'$;  $t_{\mathrm{next}} \leftarrow t + h$
2: $\theta' \leftarrow \mathrm{EMA}(\theta)$ if *use_ema* else $\theta$
3: $\ell_1 \leftarrow \theta'(x_t, t; h')$;     $u_1 \leftarrow \mathrm{Vel}(\mathrm{softmax}(\ell_1), x_t, h', t)$;     $x^{(1)} \leftarrow \mathrm{Jump}(x_t, u_1, h')$
4: $\ell_2 \leftarrow \theta'(x^{(1)}, t_{\mathrm{mid}}; h')$;   $u_2 \leftarrow \mathrm{Vel}(\mathrm{softmax}(\ell_2), x^{(1)}, h', t_{\mathrm{mid}})$;   $x^{(2)} \leftarrow \mathrm{Jump}(x^{(1)}, u_2, h')$
5: $\ell_3 \leftarrow \theta'(x^{(2)}, t_{\mathrm{mid}}; h')$;   $u_3 \leftarrow \mathrm{Vel}(\mathrm{softmax}(\ell_3), x^{(2)}, h', t_{\mathrm{mid}})$;   $x^{(3)} \leftarrow \mathrm{Jump}(x^{(2)}, u_3, h')$
6: $\ell_4 \leftarrow \theta'(x^{(3)}, t_{\mathrm{next}}; h')$
7: **RK-4 average:** $\bar{\ell} \leftarrow \frac{1}{6}(\ell_1 + 2\ell_2 + 2\ell_3 + \ell_4)$
8: **return** $\bar{\ell}$

---

$h$ where local errors accumulate across sub-evaluations. We therefore maintain a slowly varying exponential moving average (EMA) teacher:

$$\theta' \leftarrow \beta \, \theta' + (1 - \beta) \, \theta, \qquad \beta \in [0, 1) \tag{9}$$

Stop gradients through $\theta'$. The EMA teacher provides stable, low-variance targets over $[t, t+h]$, improving convergence for large $h$ and making self-consistency training robust across step budgets.

## 4.2   CUMULATIVE SCALAR

For a CTMC token update, the marginal velocity separates into a *scale* and a *direction* (Equation (5)):

$$u_t^i(x^i, z) \;=\; \underbrace{\frac{\dot{\kappa}(t)}{1 - \kappa(t)}}_{\text{scale } g(t)} \cdot \underbrace{\left[ p_{1|t}(x^i \,|\, z) - \delta_z(x^i) \right]}_{\text{direction}}. \tag{10}$$

**Scale and jump behavior:** With a monotone scheduler $\kappa : [0, 1] \to [0, 1]$ ($\kappa(0) = 0$, $\kappa(1) = 1$), few/one-step sampling uses a large step $h$, so the first update typically occurs at small $t$; for common schedulers, this makes the instantaneous *scale* $g(t) = \dot{\kappa}(t)/(1 - \kappa(t))$ too weak to trigger moves, stalling in early steps (Appendix B.3 contains more details about this argument and our motivations). Empirically, the mean jumps per token are $\approx 1.05$ with a *uniform* source and $\approx 1.00$ with a *mask* source (most positions change at most once), so the direction term in Equation (10) is typically "spent" in a single decisive update, after which tokens rarely move. In this regime, getting the scale right for each finite step dominates quality.

> ***Key question:*** *How can we incorporate the current time t and the step budget h into the scale so that, even when t is small, a single finite step delivers the right amount of flow?*

To provide the correct amount of probability flow over a finite step, we replace the instantaneous scale by a *Cumulative Scalar* obtained by integrating $g$ over the interval and normalizing by its length:

$$G_{t,h} = \int_t^{t+h} \frac{\dot{\kappa}(\tau)}{1 - \kappa(\tau)} \, d\tau = \ln \frac{1 - \kappa(t)}{1 - \kappa(t+h)} \,, \qquad \bar{g}_{t,h} = \frac{G_{t,h}}{h} = \frac{1}{h} \ln \frac{1 - \kappa(t)}{1 - \kappa(t+h)}. \tag{11}$$

Substituting the Cumulative Scalar yields

$$\bar{u}_t^i(x^i, z) = \bar{g}_{t,h} \left( p_{1|t}(x^i \,|\, z) - \delta_z(x^i) \right), \tag{12}$$

which calibrates the step strength using both $t$ and $h$, enabling effective jumps even when $t$ is small. Using the Cumulative Scalar in Equation (12) addresses this and improves few-/one-step generation.

## 4.3   TRAINING APPROACH

We train the step-aware generator $\theta$ to be *locally faithful* to the DFM path at small steps and *globally consistent* with a shortcut teacher over large steps. Each minibatch provides $(x_t, x_1, t, h)$, where $h \in (0, 1]$ and $h + t \leq 1$ is the intended step size (the step budget). The student produces logits

$\ell = \theta(x_t, t; h)$, which are used both in a small-step *path* objective (Equation (8)) and for comparison against a large-step *teacher* defined on $[t, t+h]$. The sampling process is detailed in Appendix B.1.

**Shortcut teacher.** To stabilize and supervise large moves, we integrate $\theta$ across the interval using a shortcut scheme, implemented with an EMA copy $\theta'$ yielding averaged logits $\ell_{\text{tea}}$:

$$\ell_{\text{tea}} \;\leftarrow\; \text{RK-2 / RK-4 Estimate}\big(\theta', x_t, t, h\big)$$

The teacher is treated as a stop–gradient (no backprop through $\theta'$). After each optimizer step, we update the EMA parameters via Equation (9).

**Losses.** Let $p_\theta(\cdot \mid x_t, t, h) = \text{softmax}(\ell/T)$ and $p_{\text{tea}}(\cdot \mid x_t, t, h) = \text{softmax}(\ell_{\text{tea}}/T)$ with temperature $T \geq 1$ (in our experiments, $T$ is always set to 1). Let $L$ denote the context length. We compute losses tokenwise over all positions and average per sample.

$$\mathcal{L}_{\text{dist}} \;=\; \frac{1}{L}\sum_{j=1}^{L} D_{\text{KL}}\big(p_{\text{tea},j} \,\|\, p_{\theta,j}\big) \quad \text{(stop-grad on } \ell_{\text{tea}}\text{)}.$$

For the DFM path objective, we use the per-token loss from Equation (8) averaged across tokens:

$$\mathcal{L}_{\text{dfm}} \;=\; \frac{1}{L}\sum_{j=0}^{L} \mathcal{L}_j(x_1, x_t, t; h),$$

where (from Equation (8))

$$\mathcal{L}_j(x_1, x_t, t; h) = -\,\bar{g}_{t,h}\Big[ p_{1|t}(x_t^j \mid x_t) - \delta_{x_1^j}(x_t^j) + \big(1 - \delta_{x_1^j}(x_t^j)\big)\log p_{1|t}(x_1^j \mid x_t) \Big],$$

**Budget-aware blending.** With a threshold $\tau$ on the step size, blend per sample $b$ such that tiny steps $(h < \tau)$ optimize the DFM path loss $\mathcal{L}_{\text{dfm}}$, while larger steps distill to the shortcut teacher.

$$m_b \;=\; \mathbb{I}[\,h_b < \tau\,], \qquad \mathcal{L} \;=\; \frac{1}{B}\sum_{b=1}^{B}\Big(m_b\,\mathcal{L}_{\text{dfm}}^{(b)} \;+\; (1 - m_b)\,\mathcal{L}_{\text{dist}}^{(b)}\Big). \tag{13}$$

## 5 EXPERIMENTS

### 5.1 EXPERIMENTAL SETUP

**Training.** Step-aware DFM training incurs additional model evaluations per batch (for shortcut RK-2/RK-4 teachers), making from-scratch optimization expensive. We therefore adopt a **pretrain → fine-tune** protocol: first pretrain a plain DFM backbone, then fine-tune it with step-aware objectives and the cumulative scalar update. We pretrain the DFM model following Gat et al. (2024) and cover two source distributions—*uniform* and *mask*—and three model sizes: 0.169B, 1.3B, and 1.7B parameters for the uniform source, plus a 0.169B mask-source model. These checkpoints serve as initialization for FS-DFM fine-tuning. Appendix C.1 contains all architecture and training details.

We train on FineWeb-Edu (Lozhkov et al., 2024) and evaluate on WikiText-103 (Merity et al., 2017). We use GPT-2 tokenizer and, during preprocessing, we append an EOS token to each document and pack the resulting token stream into contiguous blocks of length $1\,024$. We concatenate shorter samples to reach the desired $1\,024$ token target.

**Step–size schedule.** Shortcut integration (RK-2/RK-4) requires evaluating the model at both $h$ and $h/2$ inside training step. To cover a broad range of inference budgets with a single model, we sample $h$ from a logarithmic grid $h \in \{2^k\}$ for $k \in \{-10, -9, -8, \ldots, -1, 0\}$. The grid covers everything from tiny steps for precise path following to very large steps for few-step generation. In each minibatch, we sample $h$ from the grid (uniform over values unless stated otherwise); the shortcut teacher then internally uses the required $h/2$ sub-evaluations. We report variants of the $h$-sampling policy and their impact in Appendix E.1. For budget-aware blending (Equation (13)), we set the threshold $\tau = 2^{-9}$ for simplicity.

**Scheduler choice** $\kappa$**.** There are many valid probability–path schedulers in DFM (e.g., convex or cosine). For simplicity, we use the linear scheduler $\kappa(t) = t$, which yields $g(t) = \dot{\kappa}(t)/(1 - \kappa(t)) =$

Table 1: Ablation on scaler formulation across NFEs: integrating the scheduler within each step (Cumulative Scalar) yields a closed-form, probability-preserving update that sharply lowers GPT-2 perplexity—especially at 1–2 NFEs—while maintaining comparable entropy.

| Solver | 1 | | 2 | | 4 | | 8 | | 1 024 | |
|---|---|---|---|---|---|---|---|---|---|---|
| | ppl. | ent. | ppl. | ent. | ppl. | ent. | ppl. | ent. | ppl. | ent. |
| Scaler | 1 312.65 | 6.45 | 462.31 | 6.42 | 194.29 | 6.90 | 97.51 | 7.16 | 85.61 | 7.84 |
| Cum. Scalar | 514.40 | 6.08 | 333.07 | 6.60 | 176.19 | 6.97 | 90.49 | 7.29 | 87.36 | 7.91 |

$1/(1-t)$ and the cumulative scalar $\bar{g}_{t,h} = \frac{1}{h}\ln\left(\frac{1-\kappa(t)}{1-\kappa(t+h)}\right) = \frac{1}{h}\ln\left(\frac{1-t}{1-t-h}\right)$. This choice keeps the path well–behaved, simplifies implementation, and focuses our study on step–aware training rather than scheduler design.

**Shortcut Model.** We make the generator *step–aware* by conditioning on the intended step size $h$ alongside time $t$, i.e., $\ell = \theta(x_t, t; h)$. During training we sample $h$, construct a shortcut teacher over $[t, t+h]$ using an EMA copy of the student, and apply the budget-aware losses from Section 4.3. During inference, we fix a budget of $S \in \{2^k\}$ steps and $k \in \{0, 1, 2, \ldots, 9, 10\}$, set $h = 1/S$, and run the step-aware sampler. Implementation details appear in Appendix C.2.

**Measurement.** We report several complementary metrics: *Perplexity (PPL)* measured by a fixed reference LM (`gpt2-large`); lower is better; and *Entropy* – the average uncertainty of our model's token distributions; lower indicates sharper, more decisive predictions; *Token accuracy*: the fraction of model prediction that match the ground-truth; higher is better; and MAUVE (Pillutla et al., 2021), divergence-based metric that measures how similar the distribution of text generated by a model is to that of real human text. All metrics are computed on the evaluation split and averaged across sequences.

**Baselines.** We compare FS-DFM to a broad set of discrete diffusion and few-step generative models. In the main text, we report results against two diffusion language models, LLaDA-8B (Nie et al., 2025b) and Dream-7B (Ye et al., 2025), each in *Base* and *Instruct* variants, as well as discrete flow-matching (DFM) baselines at three model sizes (0.169B, 1.3B, 1.7B). To further contextualize performance, Appendix E.3 expands this comparison to include multi-round refinement models such as SDTT (Deschenaux & Gulcehre, 2025), hybrid discrete–continuous diffusion models such as HDLM (Fathi et al., 2025), and several masked-diffusion / re-masking systems (MDLM, SEDD, ReMDM), all evaluated under a fixed few-step generation budget.

## 5.2 RESULTS

**Cumulative Scalar Improves Few-Step Sampling.** Table 1 shows RK-4 + Cumulative Scalar performs better than only RK-4. Consistent gains at small budgets: GPT-2 perplexity drops by **60.8%** at **1 NFE** ($1\,312.65 \to 514.40$), **28.0%** at 2, **9.3%** at 4, and **7.2%** at 8. The improvement comes from integrating the scheduler's rate over $[t, t+h]$ (Equation (11)), which better matches a single large step to the effect of many small steps and reduces discretization bias. As $h$ shrinks (more NFEs), the gap narrows but Cumulative Scalar remains superior. Entropy stays comparable and is slightly higher at larger budgets (e.g., 7.29 vs. 7.16 at 8 NFEs), indicating preserved diversity alongside lower perplexity.

**The effect of RK-4 compared to RK-2 (Heun average)** is shown in Figure 3. All evaluations use Cumulative Scalar. Overall, RK-4 performs better for generation because it delivers lower perplexity (typical/median $\simeq 80$ vs. RK-2's $\simeq 84$), especially at lower NFEs. RK-4 generated a better result with a huge cap at fewer steps, by increasing the value of steps, the cap will decrease, but still, RK-4 is better. The trade-off is a slightly higher entropy (median $\simeq 7.63$ vs. 7.50), but the perplexity improvement is larger and more relevant for text quality. Using the median to compare methods is helpful because both metrics (especially perplexity) can be skewed by occasional spikes at certain NFEs. Appendix E discusses RK-4 and RK-2's training-inference time trade-offs, includes an investigation into how step-size weights shape few-step fidelity, and the effect of source distribution.

**FS-DFM vs. DFM performance across scales** is shown in Figure 1 which compares the two methods at three parameter sizes. Across all sizes, FS-DFM reaches the strong-quality regime in far fewer function evaluations: perplexity drops sharply, accuracy saturates early, and entropy converges to the

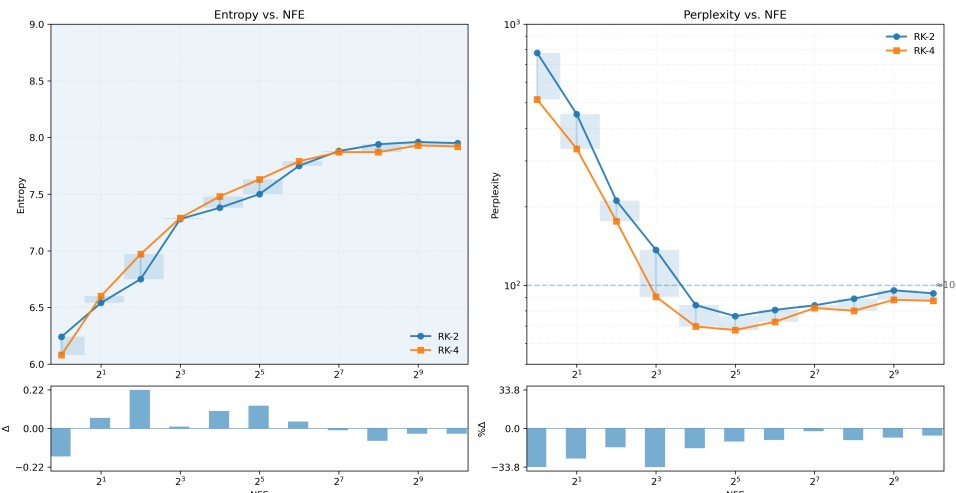

Figure 3: RK-2 vs. RK-4 across NFE. Top panels show entropy (linear y) and perplexity (log–log), with ribbons and vertical connectors highlighting pointwise gaps. The bottom shows the deltas ($\Delta$ entropy = RK-4 − RK-2; %$\Delta$ perplexity = RK-4/RK-2 −1). RK-4 has consistently lower perplexity ratio $\simeq 0.88\times$ with a small entropy trade-off (median 7.63 vs. 7.5), making it the stronger choice for generation over most NFE settings.

Table 2: **FS-DFM vs. diffusion LMs across step budgets.** Each method receives a 512-token prefix; metrics are computed on the 512-token continuation only (ppl = perplexity, ent = entropy, MVE = MAUVE Pillutla et al. (2021)). FS-DFM attains competitive quality in few steps with stable entropy across sizes, whereas baselines generally require many steps.

| Method | Size (B) | 1 ppl | ent. | MVE | 2 ppl | ent. | MVE | 4 ppl | ent. | MVE | 8 ppl | ent. | MVE | 16 ppl | ent. | MVE |
|---|---|---|---|---|---|---|---|---|---|---|---|---|---|---|---|---|
| Dream | 7.00 | 1 163.08 | 1.55 | 0.005 | 785.87 | 1.43 | 0.005 | 752.11 | 1.53 | 0.005 | 739.40 | 1.74 | 0.006 | 630.30 | 2.31 | 0.005 |
| LLaDA | 8.00 | 256.07 | 0.84 | 0.005 | 290.35 | 0.59 | 0.005 | 495.17 | 0.47 | 0.005 | 441.26 | 0.42 | 0.005 | 432.65 | 0.50 | 0.005 |
| **FS-DFM** | 0.17 | 173.39 | 7.67 | 0.006 | 143.77 | 7.85 | 0.008 | 97.07 | 7.89 | 0.053 | 75.78 | 7.95 | 0.270 | 67.42 | 7.97 | 0.390 |
| **FS-DFM** | 1.30 | 231.89 | 7.88 | 0.006 | 169.99 | 7.97 | 0.013 | 99.79 | 7.98 | 0.120 | 70.97 | 8.01 | 0.480 | 59.84 | 7.99 | 0.583 |
| **FS-DFM** | 1.70 | 191.20 | 7.67 | 0.007 | 155.01 | 7.93 | 0.014 | 101.20 | 8.03 | 0.083 | 72.84 | 8.07 | 0.311 | 61.67 | 8.06 | 0.550 |

same range as many-step DFM. The gold markers at NFE=8 highlight that FS-DFM already operates near DFM's quality plateau after only a few steps, while DFM requires substantially more steps to reach the same values. To measure accuracy across steps and sizes, we randomly change 50% of the tokens in each sequence with other tokens, provide the remaining 50% as context, and measure the ability to predict the changed tokens. FS-DFM consistently matches or surpasses DFM with dramatically fewer evaluations. This behavior is consistent between all evaluated model sizes. As NFE increases (i.e., step size $h \to 0$), FS-DFM smoothly converges to DFM: the cumulative scalar factor satisfies $\bar{g}_{t,h} \to g(t)$, so the updates coincide and the curves merge. The slight accuracy dip at very large NFEs likely reflects train–test mismatch: the model and shortcut teacher are optimized for large $h$, so many small steps accumulate discretization/renormalization bias and under-use the step-aware conditioning.

**Comparison to State-of-the-Art Diffusion LMs.** Table 2 compares FS-DFM with contemporary base diffusion LMs in the few-step regime ($1 \to 16$). Even though the largest FS-DFM model is more than four times smaller than LLaDA and Dream, the comparison illustrates FS-DFM's generation quality. FS-DFM reaches a strong-quality regime in *few steps* across model sizes: perplexity drops rapidly as steps increase while entropy remains well-behaved, indicating calibrated predictions without long iterative trajectories. In contrast, LLaDA and Dream are sensitive to step count and require many more denoising steps and do not reach comparable perplexity. Even after 16 steps, LLaDA and Dream achieve MAUVE scores equivalent to our smallest models score after a single step, despite being 41 times larger. A quick look at LLaDA and Dream's generations shows the models just repeat some tokens many times in a few-step setting (see Figure 8). Full result is available in Appendix E.4.

## 6 DISCUSSION AND CONCLUSION

We introduced FS-DFM, a step-aware discrete flow-matching language model that matches the perplexity of a similar-size 1 024-step discrete-flow baseline in 1 024 tokens generation with just 8 steps, yielding up to 128 times faster sampling. The key ideas in FS-DFM are (i) conditioning on a user-specified step budget and training such that one large move agrees with many small ones, and (ii) operating at the interval level via a cumulative scalar update that preserves the probability simplex and remains stable at large steps. We realized these ideas with shortcut teachers built inside the DFM framework and stabilized by an EMA teacher. Together, these components make few-step sampling accurate, controllable, and robust. While our experiments used the Runge-Kutta methods to approximate solutions to the Kolmogorov equations, it would be interesting to explore other ODE solvers to determine how they might affect either generation performance or training efficiency. Finally, because research in one/few-step diffusion generation is hindered by a lack of public discrete flow-matching models, we release our DFM and FS-DFM models and code. Appendix D presents more information about future work and limitations.

## ACKNOWLEDGEMENT

We would like to thank Ruixiang Zhang for his valuable insights and continued support throughout the project, particularly for his thoughtful feedback and guidance in addressing key challenges.

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

# Appendix

APPENDIX CONTENTS

# A PRELIMINARIES

## A.1 NOTATION

We set the following notations:

- $L$ denotes the length of sequences,

- $V$ denotes the *vocabulary*, i.e. the set of token ids, so $|V|$ denotes the vocabulary size,

- For a sequence $x$ and integer $1 \leq i \leq L$, $x^i$ denotes the value of $x$ at position $i$. So $x = (x^1, \ldots, x^L)$,

- For sequences $x$ and $y$ we set

$$\delta_y(x) = \prod_{i=1}^{L} \delta_{y^i}(x^i)$$

  where $\delta_{y^i}$ denotes the usual delta function,

- For a sequence $y$ and $x^i$, we set $\delta_y(x^i) = \delta_{y^i}(x^i)$,

- for sequences $x$ and $y$, we set $\delta_y(\overline{x^i}) = \delta_{\overline{y^i}}(\overline{x^i}) := \prod_{j \neq i} \delta_{y^j}(x^j)$

## A.2 BASICS OF CONTINUOUS TIME MARKOV CHAINS

We include an overview of the theory of CTMCs focusing on the time-inhomogeneous case and where the state space $S$ is discrete, since that is the type of Markov chain modeled in this paper.

A CTMC is a stochastic process $(X_t)_{0 \leq t \leq 1}$ indexed by a continuous variable $t$ that satisfies the *Markov property*: for a sequence $0 \leq t_1 < t_2 < \cdots < t_n \leq 1$ and elements $i_1, \ldots, i_n \in S$,

$$\mathbb{P}(X_{t_n} = i_n | X_{t_1} = i_1, \ldots, X_{t_{n-1}} = i_{n-1}) = \mathbb{P}(X_{t_n} = i_n | X_{t_{n-1}} = i_{n-1}).$$

The evolution of a CTMC is dictated by its *transition matrices* $P(s, t)$ for $0 \leq s \leq t \leq 1$ where for state $x$ and state $y$ in $S$,

$$P_{x,y}(s, t) = \mathbb{P}(X_t = y | X_s = x).$$

We can derive the *Chapman-Kolmogorov* equations,

$$P(s, t) = P(s, r)P(r, t) \tag{14}$$

for $s \leq r \leq t$. Note that $P(s, t)$ is a stochastic matrix.

Assuming that $P$ is differentiable in $s$ and $t$, we can define *infinitesimal generator* or *generator* as

$$u_t := \lim_{h \to 0} \frac{P(t, t + h) - I}{h}$$

where $I$ denotes the identity matrix. For states $x$ and $y$, $u_t(x, y)$ denotes the $(x, y)$-entry of $u_t$. The generator and the transition probabilities also satisfy *Kolmogorov forward equation*

$$\partial_t P(s, t) = P(s, t)u_t \tag{15}$$

and the *Kolmogorov backward equation*

$$\partial_s P(s, t) = -u_s P(s, t) \tag{16}$$

**Proposition A.1.** *Let $u_t$ be the infinitesimal generator of a CTMC, then $u_t$ satisfies the following conditions.*

1. $u_t(x, y) \geq 0$ *for all $x \neq y$ and $t$,*

2. $\sum_y u_t(x, y) = 0$ *for all $x$ and $t$,*

3. $u_t(x, x) \leq 0.$

The entries $u_t(x, y)$ are the rate of change in probability from state $x$ into $y$ when $x \neq y$, while $-u_t(x, x)$ is the rate of change in probability for staying in state $x$. In other words, we have:

$$\mathbb{P}(X_{t+h} = y | X_t = x) = \delta_x(y) + hu_t(y, x) + o(h). \tag{17}$$

That is, $u_t$ provides a first-order Taylor approximation of the transition probabilities. This suggests that we could also construct a Markov chain if we only had the instantaneous rate of change provided by the generator $u_t$.

Given a continuous generator $u_t$ that meets the conditions in Proposition A.1, one can define a Markov chain by defining $P(s, t)$ to be the unique solution to the forward and backward Kolmogorov equations. Equivalently, (assuming $u_t$ is integrable), we can write

$$P(s, t) = I + \int_s^t P(s, u)u_t du.$$

In practice, solving for $P$ is difficult, so one usually uses numerical methods to solve these ODEs.

### A.3 JUMP CHAINS

Given a CTMC over a discrete state space $S$ with generator $u_t$ we described how the transition probabilities can be derived from $u_t$ for small differences (Equation (17)). We show how this is used practically when sampling for DFM. Assume $h$ is small (for example $1/1\,024$). The *exit rate* at time $t$ for state $x$ is given by the sum

$$\lambda_x(t) := \sum_{x \neq y} u_t(x, y) = -u_t(x, x)$$

Since the state space is discrete, the stochastic processes jumps from one state to another suddenly, and is constant the rest of the time. For small $h$, we can assume that the velocity does not change (i.e. we can treat $u_t$ as constant on $[t, t + h]$). Under this approximation, the holding time until the next jump is approximately exponentially distributed with rate $\lambda_x(t)$.

In this case, the chance of jumping state can then be modeled by the Poisson distribution $1 - \exp(-h\lambda_x(t))$. In other words,

$$\mathbb{P}(\text{jump in } [t, t + h] | X_t = x) = 1 - \exp(-h\lambda_x(t)).$$

Note that taking the Taylor expansion for the RHS shows that

$$\mathbb{P}(\text{jump in } [t, t + h] | X_t = x) = h\lambda_x(t) + o(h) = \sum_{y \neq x} \delta_x(y) + hu_t(x, y) + o(h)$$

the last term being derived from Equation (17), justifying that the probability of a jump in $[t, t + h)$ is approximately exponentially distributed. Finally, if a jump occurs, the probability of transitioning from state $x$ to state $y$ is given by

$$\mathbb{P}(X_{t+h} = y | \text{jump occurs}, X_t = x) = \frac{u_t(x, y)}{\lambda_x(t)}$$

Specializing to the case of DFM, we can consider the factorized velocities $u_t^i$ for each token position $i$ (equation 2). For fixed position $i$ and state $z$. The *exit rate* is similarly defined as

$$\lambda_t^i(z) := \sum_{a \neq z_i} u_t^i(a, z) = \frac{\dot{\kappa}(t)}{1 - \kappa(t)} \Big(1 - p_{1|t}(z_i \mid z)\Big). \tag{18}$$

The jump probability in token position $i$ is also approximately exponentially distributed according to the exit rate,

$$\mathbb{P}[\text{jump in } [t, t+h)) = 1 - \exp\big(-h\,\lambda_t^i(z)\big). \tag{19}$$

Conditioned on a jump, the next token is sampled from the off-diagonals, i.e. the posterior renormalized to exclude the current token:

$$\mathbb{P}\big[X_{t+h}^i = a \mid \text{jump}, X_t = z\big] = \frac{u_t^i(a, z)}{\lambda_t^i(z)} = \frac{p_{1|t}(a \mid z)}{1 - p_{1|t}(z_i \mid z)} \quad (a \neq z_i). \tag{20}$$

# B    SUPPLEMENTARY METHODS

## B.1    SAMPLING

At test time, we simulate the *jump process* (see Appendix A.3). Choose a budget $S$ with grid $t_s$ and steps $h_s = t_{s+1} - t_s$ (uniform $h_s = 1/S$). Initialize $X_{t_0} \sim p_0$ (uniform or mask). At step $s$, compute logits $\ell_s = \theta(X_{t_s}, t_s; h_s)$ and set

$$p_{1|t_s}(\cdot \mid X_{t_s}) \; = \; \mathrm{softmax}(\ell_s/T).$$

Using the **Cumulative Scalar** $\bar{g}_{t_s,h_s}$ from Equation (11), form the per-position exit rate (cf. Equation (18))

$$\lambda_s^i \; = \; \bar{g}_{t_s,h_s}\left(1 - p_{1|t_s}\big(X_{t_s}^i \mid X_{t_s}\big)\right).$$

Draw a jump with the exponential holding-time law (Equation (19))

$$J_s^i \sim \mathrm{Bernoulli}\left(1 - e^{-h_s \lambda_s^i}\right),$$

and, if $J_s^i = 1$, sample the next token from the off-diagonals of $p_{1|t_s}(\cdot \mid X_{t_s})$ renormalized to exclude the current token:

$$X_{t_{s+1}}^i \sim \mathrm{Cat}\left(\frac{p_{1|t_s}(a \mid X_{t_s})}{1 - p_{1|t_s}(X_{t_s}^i \mid X_{t_s})} \;\Big|\; a \neq X_{t_s}^i\right), \quad \text{else set } X_{t_{s+1}}^i = X_{t_s}^i.$$

This uses exactly one forward pass of $\theta$ per step; conditioning on $h_s$ lets a single checkpoint support different budgets $S$.

## B.2    RK-2 SHORTCUT TEACHER

We include a lightweight *RK-2 (Heun)* shortcut teacher as an alternative to RK-4. It approximates the interval-averaged logits over $[t, t+h]$ with just two model evaluations: one at $(x_t, t)$ and one at a midpoint state reached by a half-step jump $(t+h/2)$. This keeps compute low (two forward passes) while still providing stable, CTMC-consistent targets for a single large step of size $h$; see Algorithm 2.

In practice, RK-2 is useful when compute or memory are tight or for ablations that isolate the effect of teacher strength. While RK-4 typically offers stronger guidance for very large steps, RK-2 strikes a good accuracy–cost balance and integrates seamlessly with our step-aware DFM setup.

---

**Algorithm 2** Shortcut RK-2 Teacher

---

**Require:**  tokens $x_t$, time $t$, step $h$; Model $\theta$; velocity Vel; CTMC jumper Jump; *use_ema* flag
  1:  $h' \leftarrow h/2$;   $t_{\mathrm{mid}} \leftarrow t + h'$
  2:  $\theta' \leftarrow \mathrm{EMA}(\theta)$ if *use_ema* else $\theta$
  3:  $\ell_1 \leftarrow \theta'(x_t, t; h')$     $u_1 \leftarrow \mathrm{Vel}(\mathrm{softmax}(\ell_1), x_t, h', t)$   $\tilde{x} \leftarrow \mathrm{Jump}(x_t, u_1, h')$
  4:  $\ell_2 \leftarrow \theta'(\tilde{x}, t_{\mathrm{mid}}; h')$
  5:  **RK-2 average:**  $\bar{\ell} \leftarrow \frac{1}{2}(\ell_1 + \ell_2)$
  6:  **return** $\bar{\ell}$

---

## B.3    CHECKERBOARD JUMP–DYNAMICS DEMONSTRATION

In this subsection, we want to motivate the cumulative scalar $\bar{g}_{t,h}$. We visualize discrete flow-matching on a synthetic $128 \times 128$ *checkerboard* target to study jump dynamics. The data generator draws the first coordinate $x_1$ uniformly from $\{0, \ldots, 127\}$; the second coordinate $x_2$ is coupled to the parity of $\lfloor x_1/32 \rfloor$, yielding alternating $32 \times 32$ blocks (checkerboard). We run a CTMC sampler on a uniform grid $t \in [0, 1]$ (100 frames) with the *instantaneous* scale $g(t) = \dot{\kappa}(t)/(1 - \kappa(t))$, and compare two source distributions $p_0$: (i) an all-[MASK] source (we add a dedicated mask token to the vocabulary), and (ii) a uniform source over tokens. We wish to emphasize that the images in Figure 4 indicate discrete flow-matching on distributions of samples (in this case points in the plane).

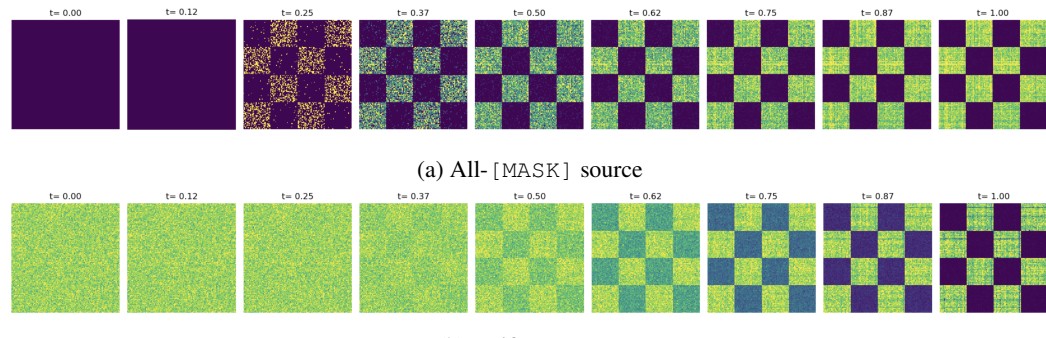

(a) All-`[MASK]` source

(b) Uniform source

Figure 4: Discrete flow-matching on a $128 \times 128$ checkerboard under the *instantaneous* scale $g(t)$. (a) With an all-`[MASK]` source, early frames exhibit almost no jumps (stalling near $t \approx 0$). (b) A uniform source jumps earlier but still under-updates at tiny $t$. Both patterns motivate the *Cumulative Scalar* $\bar{g}_{t,h}$ (Equation (11)) to supply the correct probability flow over a finite step.

The trajectories reveal the role of the scale term at early times. With an all-`[MASK]` source (Figure 4a), frames near $t \approx 0$ show almost no motion: common schedulers make $g(t)$ small at the start, so exit rates are tiny and the chain stalls despite having the correct denoising direction (see the jump-process formulation in Appendix A.3). A uniform source (Figure 4b) spreads initial mass over valid tokens and induces earlier movement, yet the first frames still under-shoot when driven solely by $g(t)$. These behaviors motivate replacing the instantaneous scale by the *Cumulative Scalar* $\bar{g}_{t,h}$ (Equation (11)), which couples the update strength to both the current time $t$ and the finite step $h$ and makes few-/one-step updates effective.

## C  IMPLEMENTATION DETAILS

### C.1  PRE-TRAIN MODEL ARCHITECTURE

**Backbone.** We use a DiT-style G (Peebles & Xie, 2023) with rotary attention. Tokens are embedded once, rotary phases are computed once per forward pass and applied to $Q, K$, and each block applies multi-head self-attention followed by an MLP. A conditioning vector $c$ drives *adaptive layer normalization*: each block predicts per-channel shift/scale (and a small residual weight) from $c$ and applies them before attention and before the MLP. The final linear head is zero-initialized for stable start-up and produces logits; conversion of logits to a CTMC generator and ODE stepping are handled by the solver outside the backbone.

**Time and step-size conditioning.** We embed continuous time $t \in [0, 1]$ with a sinusoidal MLP. We also embed the intended inference step size $h$ and fuse the two with a linear+SiLU layer:

$$c = \phi_{\text{time}}(t) \quad \text{or} \quad c = \text{SiLU}\big(W\left[\phi_{\text{time}}(t); \phi_{\Delta t}(h)\right]\big).$$

The same $c$ is fed to all transformer blocks and the final layer, making the model *step-aware*: a single set of weights works with different integrator schedules at inference (few large steps or many small ones) without retraining. The backbone's output remains logits; downstream components convert these to CTMC-valid generators and perform the PF-ODE update for the chosen step size $h$.

### C.2  SHORTCUT MODEL: IMPLEMENTATION DETAILS

**Training/evaluation protocol.** In training, the student predicts $\ell = \theta(x_t, t; h)$ and is supervised by (i) a small-step path loss and (ii) a large-step distillation loss against the shortcut teacher, blended by the budget-aware rule in Equation (13). At evaluation, we select a step budget $S$, set $h = 1/S$, and apply the Euler-with-average-velocity update each step; no additional tuning is required.

### C.3 TRAINING DETAILS

For initialization, we run 1M iterations with a global batch size of 128; each sample contributes 512 predicted tokens, for a total of 65.5B supervised tokens. For fine-tuning (same 512 predicted tokens/sample, 500K iterations per model), batch sizes are 96 (0.17B),32 (1.3B), and 16 (1.7B), yielding about 24.6B, 8.19B, and 4.10B tokens, respectively. Training used $8\times$ A100–80GB compute nodes. We optimize all models with AdamW, using a cosine-annealed learning rate schedule. We use a peak learning rate of $3\mathrm{e}{-4}$ for pre-training and $1\mathrm{e}{-5}$ for fine-tuning.

## D LIMITATIONS AND FUTURE WORK

Our study primarily targets long-horizon language modeling, and our evaluation emphasizes perplexity and masked-token accuracy; broader assessments (instruction following, QA/reasoning, and multilingual settings) remain to be done. A practical limitation is the lack of public checkpoints for discrete flow-matching: Accordingly, we plan to release DFM and FS-DFM models and training code to support reproducibility and downstream research. On efficiency, we aim to learn adaptive, position-aware step schedules and budget controllers that maintain few-step quality while further reducing inference cost. Scaling to ultra-long contexts (8–32k+ tokens) with memory-efficient attention and retrieval is another important direction. Beyond likelihood metrics, we will incorporate alignment-oriented objectives (e.g., preference optimization) to improve instruction fidelity and calibration. Another choice made in our experiments was using the Runge-Kutta method RK-4 to approximate solutions to the Kolmogorov equations. It would be interesting to experiment with other ODE solvers and see how they affect performance. Also, we only inspected either uniform source or masked sources, it would be interesting to examine how a hybrid scheduler mixing masked and uniform sources could affect performance. Finally, integrating FS-DFM with systems techniques such as speculative decoding and cache reuse could compound latency gains at deployment, helping translate few-step discrete flows into practical, general-purpose generators.

## E MORE RESULTS

This section includes more experiments, especially ablation studies, and some generated samples.

### E.1 ABLATION STUDIES

We assess each design choice by varying a single factor while holding the rest of the pipeline fixed. Unless stated otherwise, all ablations use the same data split, model size, optimizer and schedule (AdamW with cosine), the step-size grid $\{1/1024, \ldots, 1\}$ with same policy of sampling, the shortcut teacher (RK-4 with EMA) during fine-tuning, the *Cumulative Scalar* for training and sampling, a uniform source distribution, and identical inference budgets $S \in \{1, 2, 4, 8\}$ with temperature $T{=}1$.

**Effect of Step-Size Sampling Weights on Few-Step Fidelity.** We ablate the *training-time sampling weights* over the step-size grid from Section 5.1 (ordered from the smallest to the largest step) to understand how biasing minibatch selection toward large steps affects few-step performance. We evaluate five policies: (i) a "tail-boosted" **(TB-10)** scheme with uniform weights except on the largest step, $[\mathbf{10}, 1, 1, 1, 1, 1, 1, 1, 1, 1, 1]$; (ii) a stronger tail boost **(TB-20)**, $[\mathbf{20}, 1, 1, 1, 1, 1, 1, 1, 1, 1, 1]$; (iii) pure uniform **(PU)** with the same weight on all step sizes; (iv) a geometric **(G)**, large-step–biased schedule $[2^{10}, 2^9, \ldots, 2^1, 2^0]$; and (v) an *annealed geometric* **(AG)** variant that starts as geometric **(G)** and transitions to pure uniform **(PU)** by doubling weights every $10\,000$ steps up to $2^{10}$, so by $100\,000$ steps, all weights equal $2^{10}$. This experiment probes two intuitions: (a) modest to strong emphasis on the largest $h$ should sharpen single/few-step accuracy by exposing the student to more shortcut-distilled targets; and (b) annealing the bias away (policy AG) preserves this benefit early while restoring balanced coverage later, stabilizing path-following at small $h$ without sacrificing few-step quality. The results (Table 3), indicate that the strongest tail boost, **(TB-20)**, performs best. We adopt this setting for all subsequent experiments.

**Effect of source Distribution for Few-Step Generation.** We study the effect of the *source distribution* used by DFM on few-step sampling. Following Gat et al. (2024), we compare a **mask source** (all positions start as a dedicated [MASK] state) against a **uniform source** (initialized with random

Table 3: Step-size weight ablation over $h \in \{2^{-k}\}$ for $k \in \{-10, -9, -8, \ldots, -1, 0\}$: TB-10, TB-20, PU (uniform), G (geometric), AG (annealed). Bias toward large $h$ improves few-step quality.

| Policy | 1 | | 2 | | 4 | | 8 | | 1 024 | |
|---|---|---|---|---|---|---|---|---|---|---|
| | ppl. | ent. | ppl. | ent. | ppl. | ent. | ppl. | ent. | ppl. | ent. |
| **TB-10** | 834.85 | 6.51 | 325.32 | 6.66 | 159.59 | 6.99 | 94.72 | 7.19 | 85.13 | 7.94 |
| **TB-20** | 514.40 | 6.08 | 333.07 | 6.60 | 176.19 | 6.97 | 90.49 | 7.29 | 87.36 | 7.92 |
| **PU** | 3.01 | 0.64 | 30.81 | 1.73 | 18.67 | 2.31 | 21.54 | 2.73 | 262.53 | 6.96 |
| **G** | 1 353.99 | 7.57 | 635.59 | 7.71 | 250.89 | 7.85 | 116.99 | 7.85 | 96.20 | 7.95 |
| **AG** | 906.15 | 5.82 | 439.41 | 5.93 | 161.82 | 6.37 | 87.73 | 6.63 | 77.83 | 7.76 |

Table 4: Effect of source distribution on FS-DFM. *Uniform* source initialization avoids first-escape bottlenecks from `[MASK]` and delivers lower perplexity with comparable entropy at small NFEs; a *mask* source often collapses in the few-step regime.

| Source Distribution | 1 | | 2 | | 4 | | 8 | | 1 024 | |
|---|---|---|---|---|---|---|---|---|---|---|
| | ppl. | ent. | ppl. | ent. | ppl. | ent. | ppl. | ent. | ppl. | ent. |
| Mask | 4 438.49 | 8.62 | 2 844.42 | 8.64 | 1 082.07 | 8.48 | 413.43 | 8.40 | 364.61 | 8.36 |
| Uniform | 777.02 | 6.24 | 451.79 | 6.54 | 211.04 | 6.75 | 136.60 | 7.38 | 93.24 | 7.95 |

tokens). With only one or a few steps, the CTMC has limited opportunity to escape `[MASK]`; transitions concentrate on a small set of outcomes or remain at `[MASK]`, producing collapse and high perplexity. In contrast, the uniform source spreads initial mass over real tokens, so the model performs corrective refinements rather than first-escape jumps. This yields markedly lower perplexity with healthy entropy at tiny budgets and remains competitive as the budget grows. Table 4 shows that uniform source is consistently superior in the few-step regime, while the performance gap narrows with larger NFE as more transitions become available. For this experiment, we used the RK-2 method for both mask and uniform sources.

### E.2 TRAINING–INFERENCE TIME TRADE-OFFS

Shortcut teachers (RK-2/RK-4) improve few-step fidelity by querying the model at multiple abscissae within each training step, which increases per-iteration compute relative to pure DFM. In our setup, RK-2 adds roughly **33%** wall-clock time per training step, while RK-4 adds about **100%** (due to the extra forward passes and intermediate updates). We mitigate this cost in two ways: (i) we *fine-tune* from a strong pre-trained checkpoint, limiting the number of optimization steps required; and (ii) we use shortcut teachers *only during fine-tuning* to teach large-step consistency, whereas *inference* uses **very few steps** (e.g., 1–8 NFEs), yielding substantial latency and energy savings over the model's lifetime. Consistent with the ablation in Appendix E.1, the stronger tail-boost policy (**TB-20**) delivers the best quality–compute trade-off: it concentrates training on large steps—where shortcut guidance is most valuable—without degrading small-step path following. Overall, the transient training overhead is amortized by the persistent inference gains, making the net runtime favorable for deployment.

### E.3 EXPANDED BASELINE COMPARISON AND POSITIONING OF FS-DFM

To more comprehensively situate the performance of FS-DFM within the broader landscape of discrete diffusion and few-step generative models, we extend our baseline comparisons in Table 5 to include a wider set of modern architectures. This expanded set goes beyond the baselines reported in the main paper and incorporates models that vary significantly in training budgets, refinement strategies, masking mechanisms, and model size. Including these baselines is essential for establishing a more complete understanding of how FS-DFM performs relative to systems that rely on iterative refinement, large-token pre-training, or hybrid continuous–discrete generative mechanisms.

A first category consists of multi-round refinement models such as SDTT (Deschenaux & Gulcehre, 2025), which improve quality through multiple decoding rounds. Although additional rounds consistently reduce perplexity, even seven rounds of refinement remain substantially behind the performance of FS-DFM at only eight sampling steps (see Table 5). Beyond this quantitative gap, the two approaches also differ conceptually: SDTT accelerates generation by sampling from a many-

Table 5: **Expanded baseline comparison under an 8-step generation budget.** All models are evaluated with the same number of sampling steps (8). For FS-DFM, the first 65.5B tokens correspond to the DFM pre-training corpus, and the additional tokens (‡) denote fine-tuning on the FS–DFM objective. This table aggregates the most widely used discrete diffusion, masked-diffusion, and hybrid few-step baselines, allowing for a unified comparison across model size, training-token scale, and evaluation metrics. Training data metrics are provided to facilitate better comparison across methods.

| Model | Size (B) | Tokens (B) | PPL | Entropy |
|---|---|---|---|---|
| Training Data | - | - | 14.7 | 7.70 |
| *Multi-round refinement models* | | | | |
| SDTT (1 round) (Deschenaux & Gulcehre, 2025) | 0.86 | – | 612.38 | 8.38 |
| SDTT (3 rounds) (Deschenaux & Gulcehre, 2025) | 0.86 | – | 376.19 | 8.19 |
| SDTT (5 rounds) (Deschenaux & Gulcehre, 2025) | 0.86 | – | 220.81 | 8.09 |
| SDTT (7 rounds) (Deschenaux & Gulcehre, 2025) | 0.86 | – | 131.71 | 7.78 |
| *Hybrid discrete–continuous diffusion models* | | | | |
| HDLM ($\epsilon = 0.01$) (Fathi et al., 2025) | 0.16 | 524 | 279.60 | 7.54 |
| HDLM ($\gamma = 0.05$) (Fathi et al., 2025) | 0.16 | 524 | 192.74 | 8.13 |
| *Masked diffusion / discrete iterative models* | | | | |
| MDLM–OWT (Sahoo et al., 2024) | 0.17 | 262 | 837.58 | 8.47 |
| SEDD (Lou et al., 2023) | 0.32 | – | 602.09 | 8.42 |
| ReMDM–cap (Wang et al., 2025) | 0.17 | 262 | 559.74 | 8.34 |
| ReMDM–rescale (Wang et al., 2025) | 0.17 | 262 | 560.06 | 8.35 |
| ReMDM–conf (Wang et al., 2025) | 0.17 | 262 | 549.86 | 8.35 |
| ReMDM–loop (Wang et al., 2025) | 0.17 | 262 | 1358.84 | 8.41 |
| *Original Discrete Flow Matching (DFM) baselines* | | | | |
| DFM (Gat et al., 2024) | 0.17 | 65.5 | 335.64 | 8.06 |
| DFM (Gat et al., 2024) | 1.30 | 65.5 | 331.36 | 8.11 |
| DFM (Gat et al., 2024) | 1.70 | 65.5 | 444.71 | 8.18 |
| *Fast Version of DFM* | | | | |
| **FS-DFM** | 0.17 | $65.5 + 24.6^{\ddagger}$ | **90.49** | 7.29 |
| **FS-DFM** | 1.30 | $65.5 + 8.2^{\ddagger}$ | **77.21** | 7.40 |
| **FS-DFM** | 1.70 | $65.5 + 4.1^{\ddagger}$ | **73.65** | 7.61 |

step teacher and distilling its predictions, whereas FS-DFM directly learns *step-size–aware finite-step Kolmogorov dynamics* via RK-based ODE integration and a cumulative scalar inside the DFM framework. In other words, FS-DFM models how probability mass should evolve under different step budgets, rather than only compressing a fixed long-run sampler, which helps explain its stronger few-step behavior.

A second category includes hybrid diffusion–autoregressive models such as HDLM (Fathi et al., 2025). These models leverage continuous-time diffusion in combination with powerful sequence encoders and benefit from extremely large training sets—often exceeding 500B tokens. Despite this enormous training budget, their perplexity remains noticeably higher than that achieved by FS-DFM using a considerably smaller pre-training and fine-tuning corpus. This reinforces the training-token efficiency of our fast flow–based method.

We also include several discrete masked-diffusion and re-masking systems—ReMDM variants, and SEDD (Lou et al., 2023). These approaches attempt to improve sample quality by strategically masking tokens and applying multi-pass denoising schedules. Some of these baselines rely on exceptionally large training corpora, reaching into the hundreds of billions of tokens. Yet even under such favorable conditions, their perplexity and entropy remain less competitive. By contrast, FS-DFM achieves substantially stronger results with a dramatically smaller token budget, highlighting the impact of our distillation strategy and few-step training design.

Finally, we include the original DFM models from (Gat et al., 2024). These models are trained on the same 65.5B-token corpus as the backbone of our system, making them a direct and informative reference point. The results show that FS-DFM improves perplexity by more than a factor of four to five compared to these predecessors while preserving the same generation budget of eight sampling steps. This comparison isolates the contribution of our flow-distilled fast sampler from confounding factors such as model size or data scale.

Table 6: **FS-DFM vs. diffusion LMs across step budgets.** Each method receives a 512-token prefix; metrics are computed on the 512-token continuation only. FS-DFM attains competitive quality in few steps with stable entropy across sizes, whereas baselines generally require many steps.

| Method | Size (B) | 1 ppl. | ent. | MVE | 2 ppl. | ent. | MVE | 4 ppl. | ent. | MVE | 8 ppl. | ent. | MVE | 16 ppl. | ent. | MVE |
|---|---|---|---|---|---|---|---|---|---|---|---|---|---|---|---|---|
| Dream-B | 7.00 | 1 163 | 1.55 | 0.005 | 785 | 1.43 | 0.005 | 752 | 1.53 | 0.005 | 739 | 1.74 | 0.006 | 630 | 2.31 | 0.005 |
| Dream-I | 7.00 | 46 371 | 0.58 | 0.005 | 64 281 | 0.35 | 0.006 | 66 348 | 0.20 | 0.007 | 62 740 | 0.13 | 0.006 | 57 694 | 0.10 | 0.006 |
| LLaDA-B | 8.00 | 256 | 0.84 | 0.005 | 290 | 0.59 | 0.005 | 495 | 0.47 | 0.005 | 441 | 0.42 | 0.005 | 432 | 0.50 | 0.005 |
| LLaDA-I | 8.00 | 8 677 | 0.53 | 0.012 | 8 869 | 0.41 | 0.014 | 9 049 | 0.40 | 0.029 | 9 259 | 0.40 | 0.027 | 9 289 | 0.43 | 0.048 |
| **FS-DFM** | 0.17 | 173 | 7.67 | 0.006 | 143 | 7.85 | 0.008 | 97 | 7.89 | 0.053 | 75 | 7.95 | 0.270 | 67 | 7.97 | 0.390 |
| **FS-DFM** | 1.30 | 231 | 7.88 | 0.006 | 169 | 7.97 | 0.013 | 99 | 7.98 | 0.120 | 70 | 8.01 | 0.480 | 59 | 7.99 | 0.583 |
| **FS-DFM** | 1.70 | 191 | 7.67 | 0.007 | 155 | 7.93 | 0.014 | 101 | 8.03 | 0.083 | 72 | 8.07 | 0.311 | 61 | 8.06 | 0.550 |

Taken together, this expanded baseline analysis demonstrates that FS-DFM consistently matches or surpasses models that rely on deeper refinement loops, larger-scale diffusion procedures, masking strategies, or extensive training-token resources. The results highlight both the sampling efficiency and the training efficiency of our approach, positioning FS-DFM as one of the strongest few-step discrete generative models currently available.

## E.4 FULL COMPARISON

Table 6 compares FS-DFM with contemporary diffusion LMs in the few-step regime ($1 \rightarrow 16$) across both baseline *and* instruct models. Even though the largest FS-DFM model is multiple times smaller than LLaDA and Dream, the comparison illustrates FS-DFM's generation quality. For each example, we supply a 512-token prefix and evaluate only the 512-token continuation. FS-DFM reaches a strong-quality regime in *few steps* across model sizes: perplexity drops rapidly as steps increase while entropy remains well-behaved, indicating calibrated predictions without long iterative trajectories. In contrast, LLaDA and Dream are sensitive to step count and require many more denoising steps for comparable generation quality. For the Base models we just gave the 512 tokens without any instruction but we prefixed inputs to the instruction models with the prompt: `Complete the following text as a coherent passage. Do not repeat the given prefix, just continue it.` + given tokens

## E.5 SAMPLE OUTPUTS

**Overview.** Figure 5 presents a complete, high-resolution version of the qualitative result shown in the main text (Figure 2), displaying the full 8-step evolution of the sample.

We would like to acknowledge Ruixiang Zhang for providing valuable insights in our initial draft, and helping to address them.

**Corruption–recovery.** Figures 6 and 7 evaluate a 1 024-token restoration task: starting from a valid sequence, we uniformly replace half of the tokens at random and ask the model to repair the corruption. FS-DFM performs this reconstruction in **8 steps** (no teacher or micro-steps at inference), demonstrating that few steps suffice to move from heavily perturbed inputs to coherent long-form text.

**Head-to-head continuations (512-token prefix).** Figure 8 compares FS-DFM with LLaDA and Dream on next-token continuation from a shared 512-token prefix (top panel). The subsequent panels show each method's 8-step outputs. For LLaDA, we use `low_confidence` remasking with block length $\max(512/\text{step}, 32)$ and temperature 0.0. This standardized setup highlights the qualitative differences between few-step discrete sampling (FS-DFM) and iterative remasking or diffusion-style baselines.

**Trajectory visualization.** To reveal where edits concentrate over time, Figure 9 colors each token by the last step at which it changed (8 bins from start→end). Tokens that stabilize early retain an "early" hue, while late edits appear in "late" hues, making convergence and late-stage polishing patterns immediately visible.

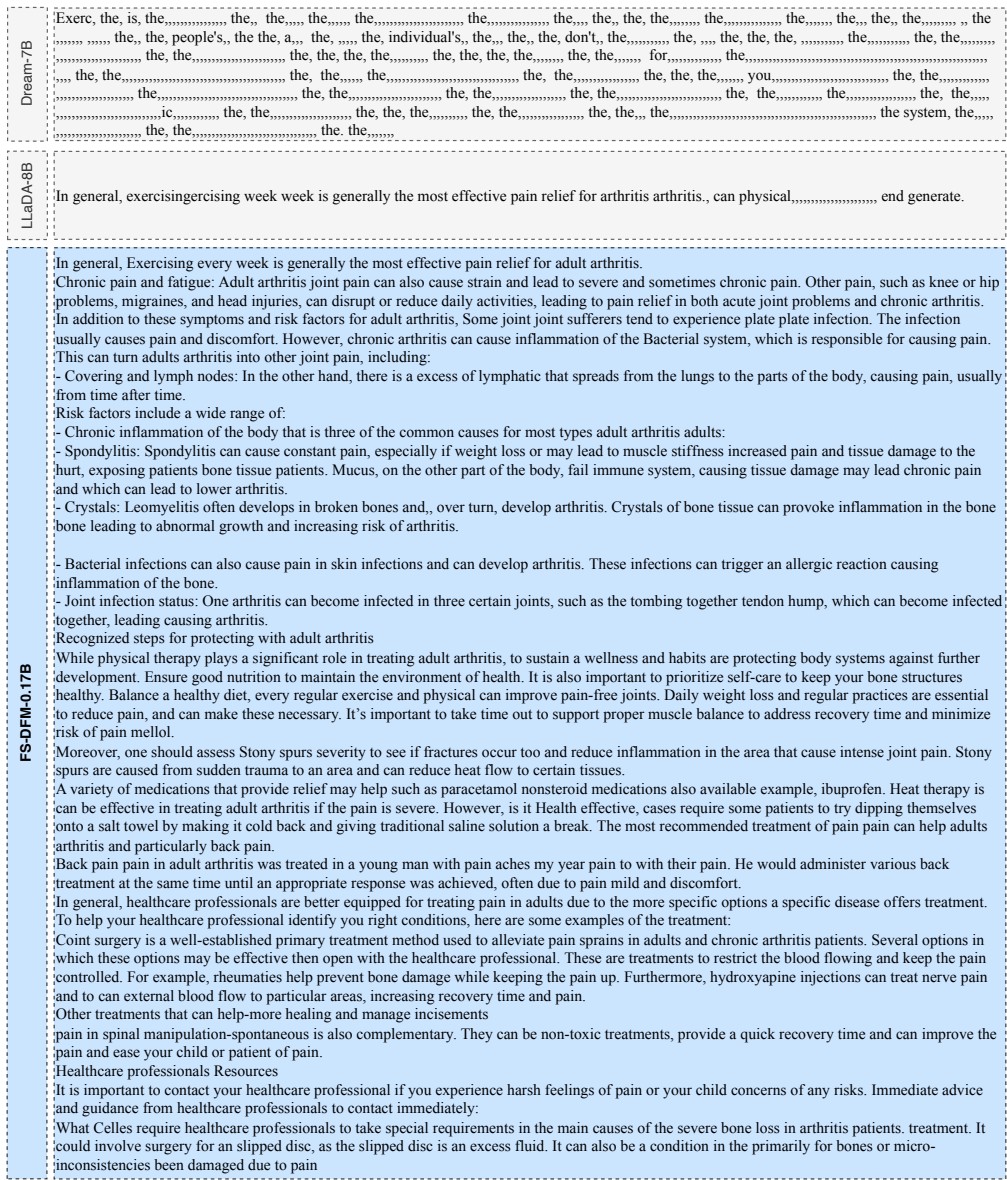

**Dream-7B**

Exerc, the, is, the,,,,,,,,,,,,, the,, the,,,,, the,,,,, the,,,,,,,,,,,,,,, the,,,,,,,,,,,,, the,,,, the,, the, the,,,,,,, the,,,,,,,,,,,,, the,,,,,, the,,, the,, the,,,,,,,,, ,, the ,,,,,, ,,,,, the,, the, people's,, the the, a,,, the, ,,,, the, individual's,, the,,, the,, the, don't,, the,,,,,,,,, the, ,,,, the, the, the, ,,,,,,,, the,,,,,,,, the, the,,,,,,, ,,,,, the, the,,,,,,,,,,,,,,,,,,,,,,, the, the, the, the, the,,,,,,,,, the, the, the, the,,,,,,, the, the, the,,,,,, for,,,,,,,,,,, the,,,,,,,,,,,,,,,,,,,,,,,,,,,,,,,,,,,,,,,,,,,,,,,,, ,,,, the, the,,,,,,,,,,,,,,,,,,,,,,,,,,,,,,,,, the, the,,,,, the,,,,,,,,,,,,,,,,,,,,,,,, the, the,,,,,,,,,,, the, the, the,,,,,, you,,,,,,,,,,,,,,,,,,,,,,, the, the,,,,,,,,,,,,,,,,,,,,,,, the, the,,,,,,,,,,,,,,,,,,,,,,, the, the,,,,,,,,,,,,,,,,,,,,,,,,,, the, the,,,,,,,,,, the, the,,,,,, the, the,,,,, ,,,,,,,,,,,,,,,,, ic,,,,,,,,,, the, the,,,,,,,,,,,,,,,,,, the, the, the,,,,,,, the, the,,,,,,,,,,,,, the, the,,, the,,,,,,,,,,,,,,,,,,,,,,,,,,,,,,,,,,,,,,,,,,,,,,,,,, the system, the,,,,, ,,,,,,,,,,,,,,,,,, the, the,,,,,,,,,,,,,,,,,,,,,,,,,,, the. the,,,,,,

**LLaDA-8B**

In general, exercisingercising week week is generally the most effective pain relief for arthritis arthritis., can physical,,,,,,,,,,,,,,,,,,,, end generate.

**FS-DFM-0.17B**

In general, Exercising every week is generally the most effective pain relief for adult arthritis.
Chronic pain and fatigue: Adult arthritis joint pain can also cause strain and lead to severe and sometimes chronic pain. Other pain, such as knee or hip problems, migraines, and head injuries, can disrupt or reduce daily activities, leading to pain relief in both acute joint problems and chronic arthritis.
In addition to these symptoms and risk factors for adult arthritis, Some joint joint sufferers tend to experience plate plate infection. The infection usually causes pain and discomfort. However, chronic arthritis can cause inflammation of the Bacterial system, which is responsible for causing pain. This can turn adults arthritis into other joint pain, including:
- Covering and lymph nodes: In the other hand, there is a excess of lymphatic that spreads from the lungs to the parts of the body, causing pain, usually from time after time.
Risk factors include a wide range of:
- Chronic inflammation of the body that is three of the common causes for most types adult arthritis adults:
- Spondylitis: Spondylitis can cause constant pain, especially if weight loss or may lead to muscle stiffness increased pain and tissue damage to the hurt, exposing patients bone tissue patients. Mucus, on the other part of the body, fail immune system, causing tissue damage may lead chronic pain and which can lead to lower arthritis.
- Crystals: Leomyelitis often develops in broken bones and,, over turn, develop arthritis. Crystals of bone tissue can provoke inflammation in the bone bone leading to abnormal growth and increasing risk of arthritis.

- Bacterial infections can also cause pain in skin infections and can develop arthritis. These infections can trigger an allergic reaction causing inflammation of the bone.
- Joint infection status: One arthritis can become infected in three certain joints, such as the tombing together tendon hump, which can become infected together, leading causing arthritis.
Recognized steps for protecting with adult arthritis
While physical therapy plays a significant role in treating adult arthritis, to sustain a wellness and habits are protecting body systems against further development. Ensure good nutrition to maintain the environment of health. It is also important to prioritize self-care to keep your bone structures healthy. Balance a healthy diet, every regular exercise and physical can improve pain-free joints. Daily weight loss and regular practices are essential to reduce pain, and can make these necessary. It's important to take time out to support proper muscle balance to address recovery time and minimize risk of pain mellol.
Moreover, one should assess Stony spurs severity to see if fractures occur too and reduce inflammation in the area that cause intense joint pain. Stony spurs are caused from sudden trauma to an area and can reduce heat flow to certain tissues.
A variety of medications that provide relief may help such as paracetamol nonsteroid medications also available example, ibuprofen. Heat therapy is can be effective in treating adult arthritis if the pain is severe. However, is it Health effective, cases require some patients to try dipping themselves onto a salt towel by making it cold back and giving traditional saline solution a break. The most recommended treatment of pain pain can help adults arthritis and particularly back pain.
Back pain pain in adult arthritis was treated in a young man with pain aches my year pain to with their pain. He would administer various back treatment at the same time until an appropriate response was achieved, often due to pain mild and discomfort.
In general, healthcare professionals are better equipped for treating pain in adults due to the more specific options a specific disease offers treatment. To help your healthcare professional identify you right conditions, here are some examples of the treatment:
Coint surgery is a well-established primary treatment method used to alleviate pain sprains in adults and chronic arthritis patients. Several options in which these options may be effective then open with the healthcare professional. These are treatments to restrict the blood flowing and keep the pain controlled. For example, rheumaties help prevent bone damage while keeping the pain up. Furthermore, hydroxyapine injections can treat nerve pain and to can external blood flow to particular areas, increasing recovery time and pain.
Other treatments that can help-more healing and manage incisements
pain in spinal manipulation-spontaneous is also complementary. They can be non-toxic treatments, provide a quick recovery time and can improve the pain and ease your child or patient of pain.
Healthcare professionals Resources
It is important to contact your healthcare professional if you experience harsh feelings of pain or your child concerns of any risks. Immediate advice and guidance from healthcare professionals to contact immediately:
What Celles require healthcare professionals to take special requirements in the main causes of the severe bone loss in arthritis patients. treatment. It could involve surgery for an slipped disc, as the slipped disc is an excess fluid. It can also be a condition in the primarily for bones or micro-inconsistencies been damaged due to pain

Figure 5: 1 024-token unconditional generation in *8 sampling steps*. FS-DFM (0.17B) successfully produces 1 024 tokens under the 8-step constraint. Despite having 40x more parameters, LLaDA-8B-Instruct and Dream-7B-Instruct's 8-steps generations exhibit trailing blanks and punctuation artifacts (e.g., repeated commas). Generations are truncated. Complete version of Figure 2.

**Input**

atives faced<R> at the Association Ground in Brisbane.<R> 8000 spectators<R> the New Zealand<R> overwhelm the<R>ers<R> win 22<R> 0<R><R><R> did<R> exert themselves<R> the win,<R><R> score<R> not reflect their dominance. After<R><R> two matches, against Toowo<R><R><R> Ips<R><R>both<R> whom<R> comfortably<R><R> the<R><R> to<R><R><R><R> with Queensland<R><R><R><R><R><R>, the<R><R>half was a close affair<R> and<R> two sides<R><R> at<R> conclusion of the half.<R><R>br<R><R><R><R><R> the<R><R> and had<R><R><R><R> the<R><R><R>. Following<R><R><R> Warbrick<R> the<R><R> the<R><R> improved and<R> recovered<R><R><R><R><R>.<R><R> after<R><R> concluded<R> circulated that<R><R><R> players had been<R> <R><R> 50 by book<R> to throw the<R>. Ey<R><R> said:
<R><R> was on<R> occasion<R> this<R> that<R> of<R><R> were thought<R> in<R><R>lance,<R><R> playing<R><R>iff<R> and that<R> had<R> got<R><R><R><R>; at all<R><R><R><R><R> it at<R>.<R><R> cautioned<R> they<R> a<R> game<R><R> second half<R>
<R> response from<R><R><R><R><R> to suspend four<R><R><R><R> travelled<R><R>owoomb<R><R> where they<R> the locals<R><R> 0. The<R>atives<R><R><R><R> for only the second time<R><R><R><R>man<R><R><R><R><R> the playing<R><R><R><R>.<R> team<R> travelled back to New Zealand, and<R><R><R><R>carg<R> on 5<R><R>
<R> ==<R> to New Zealand ==
<R><R> days<R><R><R><R><R><R><R> faced South<R><R><R><R><R><R> 1<R> front<R><R> crowd of<R>,<R>. The side<R> further injury, to<R><R><R> and<R> South<R><R><R><R><R> for<R> against<R>ura<R><R> 8<R><R> Despite<R> the match<R> players down, the<R>atives<R><R> Mataura<R><R><R>. Following the<R><R><R><R> New<R><R><R> Ot<R><R> Football<R> demanded that the<R><R> management<R><R><R> levelled<R> them<R> Queensland. Eyton responded by insisting<R><R> players had only<R> suspended<R> an<R> was conducted<R> and<R> the management<R><R><R><R><R>. The<R> Rugby Union<R>since renamed<R> Union<R><R>ised<R> incident<R> aftermath in<R><R><R> Rugby Union<R>:
<R> it<R><R><R><R><R> of the game that something was wrong with the Ma<R><R> as they<R> not showing their<R><R><R> combination. Four<R><R><R><R> were<R>,<R> charge<R> made against them of<R> the<R> the<R><R><R><R><R><R><R><R> Otago<R>, who passed the following<R><R> That<R><R> heard all<R> evidence regarding the charges<R> certain<R><R><R> Native Team, and<R> received an explicit denial<R> charges<R><R><R> and<R> satisfactory explanation<R><R><R>, we are<R> opinion that<R> are<R> facts before us justifying the<R><R><R><R> is unlikely,<R><R> attitude of the Otago<R><R><R> the<R>atives before<R> departure<R><R> they<R> have dismissed the allegations if<R>inating evidence had<R><R><R><R> was back<R><R> strength following the return<R><R><R> players when<R> side<R> Otago in<R>edin.<R> N<R> out<R>ored their<R><R><R> two, and won 11 – 8<R> The<R>'s star<R> and half<R>back, Keogh,<R> in<R>edin when the team<R> for<R>church<R><R><R><R> Hawke's<R>,<R> were<R> in<R><R><R><R><R><R> 13 –<R>.<R> <R>atives<R>play<R> praised<R> The Press:<R><R><R><R> the black<R><R> remarkable<R> and<R>ness between their legs<R><R> their shoulders<R><R> their<R> and with their<R><R> side then<R> Canterbury<R><R><R>,<R> they<R>ashed 15 – 0.<R><R><R><R><R><R> <R><R><R> N<R><R> performance<R>
<R> The<R> showed<R> Saturday<R> was a<R><R> of what<R><R><R><R> practice will<R> ... it must be<R> they<R> far and away<R> <R><R><R> local men.<R> the<R>,<R><R><R><R>, dribbling,<R><R> coll<R>,<R> running they<R> very much<R> Canterbury<R> superior. <R><R> as were<R><R><R> at<R><R>,<R><R>igan<R><R>age<R> and W.<R>yard<R> the<R> of<R><R> Wyn<R><R><R>. Warbrick, and all the<R> as<R><R><R> forwards<R> the rushes of<R><R>br<R>, May<R><R><R>are, Tai<R>a<R> and Rene<R><R> the dodging and fending<R><R><R><R> every one,<R><R><R><R> opponents ...
<R><R><R><R>church<R> travelled<R><R><R> played<R><R>arapa<R><R>.<R> match was won<R> – 8,<R>

**FS-DFM-0.17B**

atives faced South at the Association Ground in Brisbane.
8000 spectators on the New Zealand team overwhelm the Queenslanders to win 22 – 0. The Natives did not exert themselves against the win, and the score did not reflect their dominance. After each of two matches, against Toowoomb and Wild Ipsiss (both of whom could comfortably play), the team continued to play to win together with Queensland. For the spectators from each side, the one-half was a close affair, and the two sides saw it at the conclusion of the half. The Warbrick had both had won the match, and had done so by playing the play over it. Following this, the Warbrick saw the score of the twoers improved and the recovered to the centre to recover. However, after the match concluded they circulated that all of the players had been getting 3 50 by book due to throw the game. Eyton himself said:
… it was on the occasion of this, that all of the people were thought, in their parlance, they were playing upon Moziff, and that they had never got to deal with training; at all it was gathered to of it at one-half, cautioned before they had a little game in the second half.
<|endoftext|> key response from the Warbrick was to suspend four, and the teams travelled to Toowoombhey, where they won the locals 2 – 0. The Natives played for the match for only the second time, so theyman busy missing it due to the playing only one Rafael away. The team then travelled back to New Zealand, and won the side-cargue on 5 July.
Several<|endoftext|> == returned to New Zealand ==
<|endoftext|> A few days after the match, the Natives faced South match 3 in Depodyi strategy 1 in front of a crowd of 1,000. The side suffered further injury, to the 16 teams and the South balls. The side went for the match against Mataura on the 8 July. Despite placing the match 3 players down, the Natives returned to Mataura with extreme impunity. Following the evidence mistakenly brought from New Zealand, the Otago World Football Council demanded that the Native team management must have it levelled against them against Queensland. Eyton responded by insisting that the players had only been suspended before an attack was conducted, and that the management must not receive any action again. The Western Rugby Union (since renamed Western Rugby Union) prioritised the incident as aftermath in Approach from Western Rugby Union accusations:
<|endoftext|> And it was clear at the beginning of the game that something was wrong with the Maura, as they were not showing their behaviour at their combination. Four days after the players were meet, a charge was made against them of stopping the play the side managed to meet, handled by the Otago Team, who passed the following question: That, having heard all the evidence regarding the charges, certain against the three Native Team, and had received an explicit denial of charges to be retained, and a satisfactory explanation of the charges, we are of opinion that there are certain facts before us justifying the inevitable charges.<|endoftext|> it is unlikely, for the attitude of the Otago Team remained against the Natives before their departure, and they could have dismissed the allegations if incriminating evidence had passed. The team was back to full strength following the return of the two players when the side left Otago in Dunedin. The Natives outscored their Wales team between the two, and won 11 – 8. The team's star, and half-back, Keogh, was in Dunedin when the team left for Newchurch. The victory of Hawke's star, who were between Canterbury in Sasaki, in Newchurch, won 13 – 0. The Natives-play was praised throughout The Press: as they dang, the black line was remarkable, and firmness between their legs, on their shoulders, inside their tail and with their arms. The side then sent Canterbury to Newchurch, and they glashed 15 – 0. When the team achieved full strength, the two Natives asserted performance:
- The victory showed the Saturday Higgins was a striking example of what they used to do in practice will moved ... it must be that they emerged far and away from the most sophisticated local men. Throughout the action, they performed speed jumping, dribbling, and running coll yards, and running they differed very much from Canterbury's superior. Further honour as were the fastest of the legs at Newchurch, Tocigan, stoppedage, and W. Paperyard on the head of Michael War Wynick and S. Warbrick, and all the team as well as transferred forwards with the rushes of Michael Warbrick, Mayneterumbare, Tailya, and Rene, performed the dodging and fending of 2022 and every one, their women and their opponents ...
- They played at Newchurch 16 travelled20 and they played at Tarapa and Eldaki. The match was won 11 – 8, and

Figure 6: Corruption–recovery on a 1 024-token passage with **50%** random replacements (<R>): FS-DFM restores coherence in **8** jump steps using the Cumulative Scalar–driven CTMC sampler.

**Input**

<R><R> corridor<R><R> in the<R><R> later<R> the<R>states.<R> was also<R><R> the<R><R> of<R> System<R> Interstate<R>ways<R><R> Additional<R>es at<R><R> Design<R><R> September<R>, or<R><R><R> the cover color<R><R> was released<R> 1955<R><R><R> government<R>ied plans for the<R><R><R><R> proposed number<R> 1958 was Interstate<R><R>I - 75). <R><R> first section<R><R><R> built in late 1957<R> early 1958 between Evergreen<R>ores<R> M<R><R><R> north<R><R>. Ignace<R> The Mack<R>ac<R> was opened<R><R> on<R> 1<R> 1957<R> a<R><R> freeway<R><R> interchange<R><R><R><R><R><R><R>. In<R><R> another new freeway<R> closed the gap between<R><R>in<R> Bridge and<R>green Sh<R><R><R><R><R>,<R><R> - 75 designation suppl<R><R> on the bridge,<R> US<R><R> shifted<R><R><R><R> 75<R><R><R> in the<R>.<R>ace<R><R><R> former<R> of<R> <R><R><R><R><R> Ignace was<R>ated BL I<R><R> 75<R><R> sections of<R> were<R> in<R> immediately to the south<R><R><R> of<R> International<R><R> Sault Ste.<R> as well<R> between<R><R><R> Kinross<R> The<R> two sections<R> in 1963 connected the northern<R> of the<R><R> M<R><R><R> Kin<R>, and the section<R> D<R> and Sault<R><R><R><R><R><R> of<R><R>'s former<R> became<R><R><R><R> as Mack<R><R> Trail (H<R><R>63).<R> The<R><R> State High<R><R><R> 2<R><R> 41 into<R> expressway between<R><R> and<R><R><R> 1971.<R><R> built a new bridge over the<R><R>ique River in<R><R> bypass<R><R><R> MD<R> disposed<R><R> former routing<R><R><R> into<R><R><R><R><R> western half was initially<R> unnumbered<R><R> until it was<R> transferred to local<R><R><R> extension<R><R> -<R><R> remainder, including<R> S<R>on Bridge,<R> downtown. In<R> same<R>,<R> <R> trunc<R> US<R> to end<R> St.<R><R> by removing<R><R> the<R><R> 75 freeway. The last<R> made to<R><R><R><R> through<R> <R><R> 1998<R><R>ing the bridge that<R><R> the<R> over the river in<R>. In 2011<R><R>OT<R> the speed limit<R> the expressway<R> in<R> County<R> 55 to<R><R> (<R><R><R><R><R><R>),<R><R> speed limit<R><R> remains 55 mph<R>89 km / h<R> <R><R> ==<R><R>ations and tourist routes ==<R><R><R> July<R>,<R>,<R> State<R><R> named<R><R><R>,<R><R><R><R><R> in<R> <R><R> B<R> Highway to honor Frank<R><R> Bohn<R><R><R> local citizen<R> later served in<R> from 1927 to<R><R><R><R>,<R> of<R>an<R><R> a<R> to<R><R> of<R><R> I called Memory Lane. The<R> consisted of elm and maple trees<R><R> US<R><R> US<R><R> <R> town<R> The<R> Legion<R> the<R> to<R> businesses and<R> who<R><R><R><R> Later<R> 1949<R><R> Bessemer<R>'s<R> created<R><R><R><R> of a<R><R> memorial to the<R> veterans<R> Also<R> Memory<R><R> the group<R> el<R> and 1,<R><R> <R><R><R> trees and<R><R><R> landscaped<R><R> along 2.3<R><R><R>.7<R><R> US<R><R> of<R><R>.<R><R> of US 2<R> along with<R><R><R> the<R> Peninsula<R><R><R><R> United<R><R> Veterans<R><R> in<R><R> To connect the gap in<R><R> where<R> 2<R><R> Wisconsin<R> M<R><R> and M<R><R><R> used in place<R> US 2 between<R> Mountain<R> Crystal Falls.<R><R> the highway were not<R> until<R> when Governor George W<R> Romney<R><R> installed<R> <R> The Amvets<R> Drive<R><R> created for the section<R><R> 2 / US<R> M - 35<R><R><R><R><R><R> limits<R><R> Road 426<R>CR 426<R><R> Delta<R><R><R> American<R><R><R>VETS)<R><R><R> petitioned the<R> Legislature<R> grant this<R>,<R> was<R> under Public Act<R><R><R><R><R><R><R><R> 2<R> part of<R> overall<R> Lakes<R> Tour<R>GLCT):<R><R><R> the<R> state line<R> Iron<R><R><R> M <R><R> junction in Wakefield<R><R><R> the<R><R> Circle<R><R>LS<R> and<R><R><R><R> southern<R> -<R> junction<R><R>anaba to<R><R> terminus in<R><R><R>ace<R> part<R> the<R> Michigan<R> Tour (<R>CT<R> These<R> tours were created in May<R><R> a joint<R><R> MDOT and its<R> in<R><R> and<R>.<R> section of US<R> between Iron<R> and<R> Falls<R> been<R><R><R> County

**FS-DFM-0.17B**

ary transport corridor that began in the 1930s later replaced the interstates. It was also involved in the developing development of a System of Interstate Railways and promoting Additional Routes at the New Design. In September 1954, or Article 26 of the cover colorat bridge was released in 1955, the federal government prophesied plans for the bridge. The first proposed number in 1958 was Interstate rail (I - 75).
- The first section of roadway was built in late 1957 or early 1958 between Evergreen Shores and M 47 just north of St. Ignace. The Mackinac Bridge was opened at first on February 1, 1957 until a pyramid-shaped freeway found the interchange and replaced by World War II. In 1957, another new freeway built closed the gap between the Mackinac Bridge and Evergreen Shores. In mid- 1957, the I - 75 designation suppl an agreed format on the bridge, but USWD eventually shifted it to M - 75. It apially in the St. Ignace segment and the former section of the bridge south to St. Ignace was demated BL I - 75. Two sections of roadway were created in 1963 immediately to the south-east intersection of International City of Sault Ste. Marie as well as between Dross and Kinross. The first two sections built in 1963 connected the northern section of the bridge between M - 75 and Kinross, and the section between Dross and Sault Ste. Marie to Washington. The lower section of the bridge's former route became part known in time as Mackinac Trail (HW63).
<|endoftext|> The Florida Blue State Highways replaced US 2 and US 41 into the expressway between Dross and W VA in 1971. US68 built a new bridge over the Mackinique River in 1971 to bypass the bridge. MDOT disposed of its former routing of the bridge into St. Ignace. The western half was initially left unnumbered in 1963 until it was later transferred to local residents as an extension of I - 75. The remainder, including the Susson Bridge, was downtown. In the same year, MDOT trunc completed US 41 to end the St. Ignace by removing it from the I - 75 freeway. The last attempt was made to extend the expressway through Clark County in 1998 bymissioning the bridge that originally marked the bridge over the river in 2009. In 2011, MDOT reduced the speed limit on the expressway traffic in Clark County from 55 to 55 mph (67 km / h) max), and the speed limit on bridge remains 55 mph (89 km / h).
- Colors ==, elevations and tourist routes ==ed<|endoftext|>In on July 15, 1933, the State Legion established named M - 1870, and God Almighty American followed in distance to the Bohn Highway to honor Frank S. Bohn, a customized local citizen who later served in politics from 1927 to 1933. In 1935, the town of Yukanaba opened a bridge to the Museum of Historic Land I called Memory Lane. The bridge consisted of elm and maple trees, from US71 and US 247 from the town. The State Legion provided the bridge to several businesses and others who owned the State Legion. Later in 1949, the Bessemer Wisconsin's she created and designed for the construction of a monument or memorial to the war veterans. Also named Memory Lane, the group consisted of elm and 1,000 elm and maple trees and Japanese Japanese. Indians landscaped plantings along 2.3 meters (4.7 meters) of US fifty instead of the stream bed.
<|endoftext|> Any of US 2, along with the bridge of the Michigan Peninsula Highway, on the United States Civil Veterans Drive Center in 1939. To connect the gap in the road where US 2 / US Wisconsin / M - 35 and M - 35 were used in place of US 2 between Iron Mountain and Crystal Falls. Bridges to the highway were not installed until 1939 when Governor George W. Romney completed were installed.
- The Amvets Minor Drive Center was created for the section of US 2 / US Wisconsin / M - 35, as part of the continued work limits for Michigan Road 426 (CR 426). The Delta State Bank for American education Training (VETS) of MDOT petitioned the state Legislature to grant this assistance, which was created under Public Act, 1939. In addition, the section of US 2 was part of the overall Great Lakes Road Tour (GLCT): This bridge connects the southern state line between Iron Mountain and southern M - 35 junction in Wakefield State 73 to the east termin Circle H1LSick, and this bridge connects the southern M - 35 junction in Yukanaba to the east terminus in St. Ignace as part of the overall Michigan Road Tour (GLCT): These road tours were created in May 1939 as a joint project between MDOT and its office in Canadian Water Resources and Forestry. The section of US 2 between Iron Mountain and Crystal Falls had been cut at Clark County

Figure 7: Corruption–recovery on a 1024-token passage with **50%** random replacements (<R>): FS-DFM restores coherence in **8** jump steps using the Cumulative Scalar–driven CTMC sampler.

| | |
|---|---|
| Input | exploration has continued at the Cordero Mine in the 21st century, and waste containing mercury and arsenic was returned there from the community of McDermitt as part of a U.S. Environmental Protection Agency cleanup project. The entire mountain range is very remote; as a result, there are few visitors, and the range offers a wilderness-like experience. Camping, hunting, fishing, and hiking are the most popular activities. The only developed recreation site nearby is at Willow Creek Hot Springs, just south of the Whitehorse Ranch, where nearby there are miles of trails designated for four-wheel off-road vehicles. Hunters come to the mountains seeking trophy mule deer, pronghorn, chukars, and rabbits. Fishing on some streams is sometimes permitted on a catch-and-release basis. The mountains are also suitable for hiking cross-country or on game trails in natural corridors along canyons and creek bottoms. There are more than 100 archaeological sites in the range that document use by Northern Paiute people as long as 7,000 years ago. Cattle grazing in the Trout Creek Mountains began in the late 19th century, and the BLM currently oversees grazing allotments in the area. Cattle can be found grazing in some areas during the spring and summer months. The effects of grazing on the local environment were the subject of controversy in the 1980s. == Land-management compromise == By the 1970s and 1980s, a century of intense cattle grazing had reduced much of the riparian vegetation along stream banks in the Trout Creek Mountains and elsewhere in the Great Basin. As a result, stream banks were eroding and upland vegetation was encroaching into riparian zones. Aspen populations declined as grazing cattle eliminated young trees, decreasing shade over streams and raising water temperatures. These conditions put the rare Lahontan cutthroat trout population at risk. Since the Lahontan was officially designated as a threatened species, environmental groups began advocating the cancellation of grazing permits in the Trout Creek Mountains. Beginning in the early 1970s, the Bureau of Land Management identified damaged riparian zones and began projects to restore natural habitat in those areas. Approximately 20,000 willow trees were planted along streams, small dams were put together to create more pools in the streams, and fencing was added to protect riparian zones from grazing. Next, the agency sought to reform land-use plans to change grazing practices, which became a complex and controversial project. As |
| Dream-7B | for the grazing grazing grazing grazing grazing grazing grazing grazing grazing grazing grazing grazing grazing grazing grazing grazing grazing grazing grazing grazing grazing grazing grazing grazing grazing grazing grazing grazing grazing grazing grazing grazing grazing grazing grazing grazing grazing grazing grazing grazing grazing grazing grazing grazing grazing grazing the grazing grazing the the the grazing the the the the the the the the the the the the the the grazing the the grazing grazing grazing grazing grazing grazing grazing grazing grazing grazing grazing grazing grazing grazing grazing grazing grazing grazing grazing grazing grazing grazing grazing grazing grazing grazing grazing grazing grazing grazing grazing grazing grazing grazing grazing grazing grazing grazing grazing grazing grazing grazing grazing grazing grazing grazing grazing grazing grazing grazing

grazing grazing grazing grazing grazing grazing grazing grazing grazing grazing grazing grazing grazing grazing grazing grazing grazing grazing grazing grazing grazing grazing grazing grazing grazing grazing grazing grazing grazing grazing grazing grazing grazing grazing grazing grazing grazing grazing grazing grazing grazing grazing grazing grazing grazing grazing grazing grazing grazing grazing grazing grazing grazing grazing grazing grazing grazing grazing grazing grazing grazing grazing grazing grazing grazing grazing grazing grazing grazing grazing grazing grazing grazing grazing grazing grazing grazing grazing grazing grazing grazing grazing grazing grazing grazing grazing grazing grazing grazing grazing grazing grazing grazing grazing grazing grazing grazing grazing grazing grazing grazing grazing grazing grazing grazing grazing grazing grazing grazing grazing grazing grazing grazing grazing grazing grazing grazing grazing grazing grazing grazing grazing grazing grazing grazing grazing grazing grazing grazing grazing grazing grazing grazing grazing grazing grazing grazing grazing grazing grazing grazing grazing grazing grazing grazing grazing grazing grazing grazing grazing grazing grazing grazing grazing grazing grazing grazing grazing grazing grazing grazing grazing grazing grazing grazing grazing grazing grazing grazing grazing grazing grazing grazing grazing grazing grazing grazing grazing grazing grazing grazing grazing grazing grazing grazing grazing grazing grazing grazing grazing grazing grazing grazing grazing grazing grazing grazing grazing grazing grazing grazing grazing grazing grazing grazing grazing grazing grazing grazing grazing grazing grazing grazing grazing grazing grazing grazing grazing grazing grazing grazing grazing grazing grazing grazing grazing grazing grazing grazing grazing grazing grazing grazing grazing grazing grazing grazing grazing grazing grazing grazing grazing grazing grazing grazing grazing grazing grazing grazing grazing grazing grazing grazing grazing grazing grazing grazing grazing grazing grazing grazing grazing grazing grazing grazing |
| LLaDA-8B | for the grazing grazing grazing grazing grazing grazing grazing grazing grazing grazing grazing grazing grazing grazing grazing grazing grazing grazing grazing grazing grazing grazing grazing grazing grazing grazing grazing grazing grazing grazing grazing grazing the the grazing grazing grazing the grazing grazing grazing grazing, grazing grazing grazing grazing grazing grazing the the grazing,, the the the the the. grazing the the the the the the the the the the the the the the the the the the the the the the the the the. the the the the the the the the the and the the the the the the the the the the the the the, the the the the the the the the the the the generations come. |
| FS-DFM-0.17B | As environmentalists protested, the BLM expressed their concern about the destructive effects by stopping cattle grazing and hiring ranchers for full access to the land until several months after it had turned temporarily to the BLM for the grazing grazing option. But, in the last several years, due to litigation, the BLM told stream operators to go immediately to court and seek authority through a for-profit corporation. Due to such efforts, the parts of the streams sampled by non-cutter communities (NGCC) in a 1978 study that moved an estimated 15 million tons of sediment in downspout and near-channel" (notwithstanding bioturbation), meaning that rainwater normally seeping through the water was not captured. The 19-year studies in the three affected streams revealed a large amount of sediment, approximately 60,000 full bulb, inches-in-depth, consumed sediment deposited in a couple of years, form the basis for smaller sediment that has been restored stream habitat. Especially in the past 60 years, increased sediment indicated a sharp decrease in laterally inflow, and, consequently, higher amounts of erosion. - Landscapes and grazing is a major environmental factor. Approximately two-thirds of total land covered in Western Rockies, a total of around 8.5 million acres of land in the North Rocky Mountain Ski Country, Colorado, is also suitable for unmanaged cattle pasture'. In 1996- there were four cattle-ding lawsuits challenging a BLM authority to eliminate North Butte Country for unmanaged pasture. The grazing threat led to the closure of approximately 95% of ski courses situated within BLM lands. While the BLM recognized those grazing rights, the removal of the land had dramatic impact on the ecosystem of the surrounding valleys and mountains. - BLM obtained water rights to the Colorado Rivers and the Walker River. Both rivers flowing south were causing the mountains to be at water levels and record-breaking low water levels in parts of the Colorado River Counties, draining almost agricultural land into minor south-central rivers. BLM's water allocations were insufficient for collapse of the Rocky Mountain Ski Region, and as a result was subject to large-scale watershed erosion. The BLM's Baseline Best Management Practice (BLM) proposes practices at managing, planning and competing sustainably for limited water supply. To make good decisions, BLM has no 100 percent specific information but generally uses different tools associated with resources and valley planning, sometimes utilizing these Water Resource Management (WRM) methods and does not forecast or consider predicting water. The current BLM prai, |

Figure 8: Compares FS-DFM with LLaDA and Dream on next-token continuation from a shared 512-token prefix. FS-DFM at 170 M parameters can generate coherent English on the provided topic in 8 steps while LLaDA and Dream fail to generate coherent English despite having 40x more parameters.

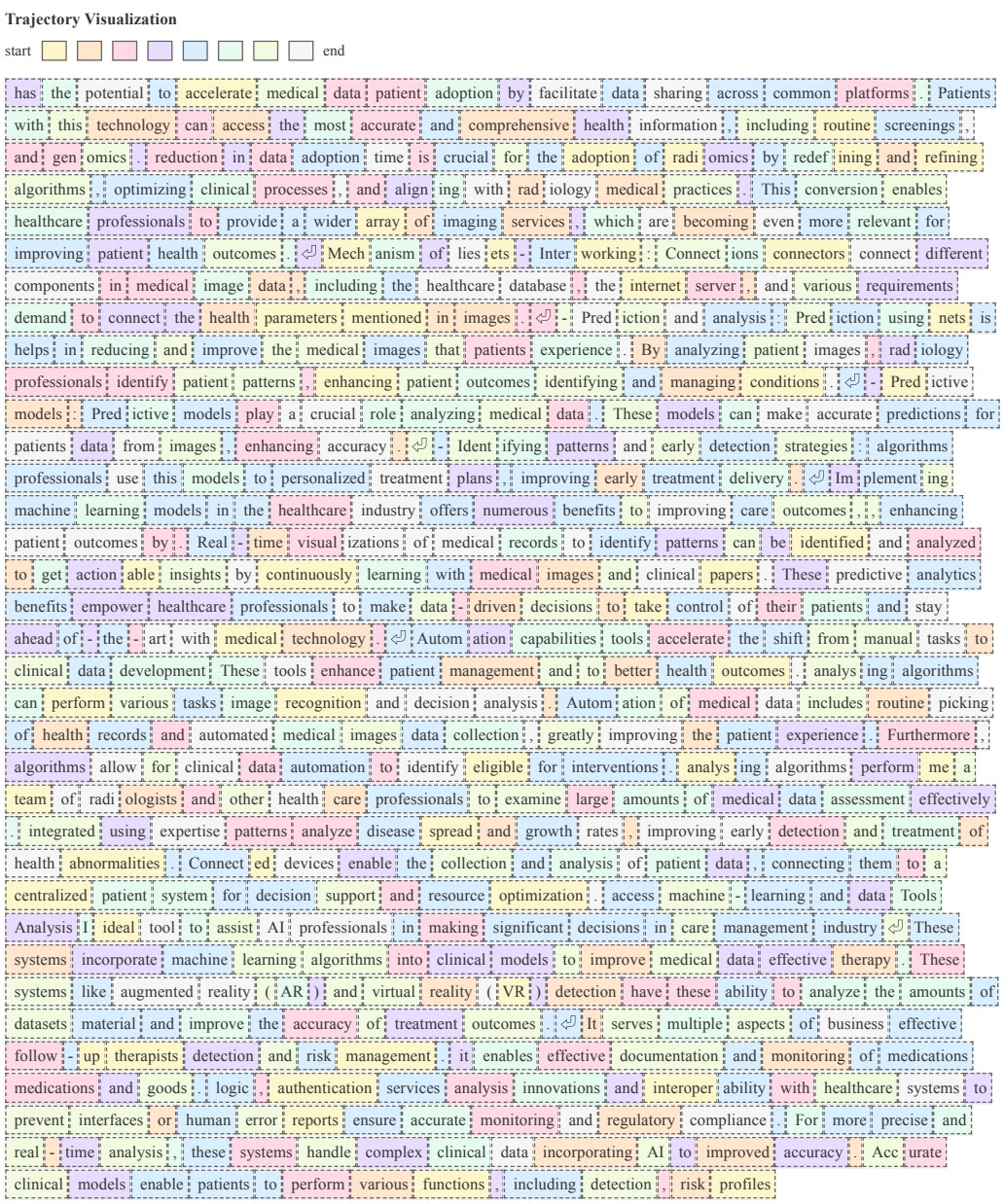

Figure 9: **Token-level generation timeline.** The displayed text is the final sample; the background of each token encodes the step of its *last change* using eight light colors (start → end). Early-stabilized tokens appear in early hues, while late edits trend toward end hues, making localized refinements and overall convergence easy to see. Note that many tokens are colored yellow, indicating they were predicted early in the process. This is due to the cumulative scalar (contrast with Figure 4).

