# OpenReview forum: "FS-DFM: Fast and Accurate Long Text Generation with Few-Step Diffusion Language Models"
_ICLR.cc/2026/Conference — ICLR 2026 Poster_

### Official Review · Reviewer_C2qF · 2025-10-20

**Soundness:** 3
**Presentation:** 3
**Contribution:** 3
**Rating:** 6
**Confidence:** 2

**Summary:**

The paper introduces FS-DFM (Few-Step Discrete Flow-Matching), a discrete diffusion language model trained to work across different step budgets. With step-aware training (conditioning on the intended step size and a shortcut-consistency objective integrated with RK4) and a Cumulative Scalar that average the flow rate over [t,t+h], the method reaches the 1024-step discrete-flow baseline quality using 8 sampling steps for 1024-token sequences. Experiments on FineWeb-Edu (training) and WikiText-103 (evaluation) across 0.17B, 1.3B, 1.7B parameters show that FS-DFM preserves quality with far fewer steps than standard discrete flows.

**Strengths:**

1. The derivation and use of the Cumulative Scalar (Eq. 12) are well motivated for finite-step transport in discrete flows. The closed-form log-ratio provides a stable update for large h and aligns with the few/one-step goal.
2. Empirically, the method matches the 1024-step baseline in 8 steps, which is a clear sampling efficiency improvement at long sequence length (1024 tokens).
3. The gains appear across three model sizes (0.17B, 1.3B, 1.7B), which suggests the effect is not tied to a single scale or architecture variant within the paper’s setup.
4. A single model supports multiple step counts through explicit conditioning, so one does not need separate models for different latency budgets.

**Weaknesses:**

1. Missing comparisons to similarly sized autoregressive baselines on the same data; this limits how readers relate the results to standard text generation references.
2. No evidence the method scales to modern 7B+ model sizes where training costs become critical.
3. Training is on FineWeb-Edu and evaluation is on WikiText-103 only; results on broader corpora or mixed-domain evaluations are not reported.
4. The evaluation focuses on open-ended continuation; there are no results on downstream or conditional tasks (for example instruction following or machine translation), so generality is uncertain.
5. All experiments use 1024 tokens (and 512→512 continuations)—unclear if advantages persist for shorter or longer sequences.

**Questions:**

1. Does the method scale beyond 1.7B parameters? If you have partial results or ablations at 7B+, please share some metrics to clarify feasibility at that scale.
2. What are the actual training costs for pretraining and the step-aware fine-tuning? It would help to report wall-clock times, hardware (GPU type and count), or an estimate in FLOPs, and how it compares to plain DFM.

---

> ### Author Response · Authors · 2025-11-22
>
> ## **W1: Autoregressive:**
>
> Autoregressive models perform one NFE for each token. Our experiments show the quality of generated text in the regime of 1 to 16 NFE include results from state-of-the-art diffusion models in Tables 2, 5, and 6.
>
>
>
> ## **W2: Lack of Scaling Evidence:**
>
> While scaling was not the focus of our work, our experiments show the scaling behavior within one order of magnitude (0.17B to 1.7). While exploring scaling up to 7B and beyond is interesting, we believe it deserves a dedicated investigation and paper.
>
>
> ## **W3: Mixed-domain evaluations:**
>
> Despite WikiText and Fineweb-edu being distantly related cousins, we believe the evaluation to be illustrative of FS-DFM’s text generation abilities, especially given the poor performance of 4x larger diffusion state-of-the art models (e.g., LLaDA and Dream). Tables 2, 5, and 6
>
>
> ## **W4: Downstream and Conditional tasks:**
>
> Small diffusion LLMs are not expected to perform well in downstream tasks. Similarly to recent <2B small-scale diffusion LLM work (SEDD, HDLM, pure-ReMDM) we focused on open-ended and conditional text generation. We extended the paper to include comparative generation experiments with those models in Table 5. We believe task performance will be an interesting comparison at scales similar to those of LLaDA and Dream.
>
>
>
> ## **W5: Limit to 1024 context length:**
>
> We agree that experiments with different sized prefixes could be interesting but we believe the experiments in Tables 2, 5, and 6 are revealing of FS-DFM’s advantage in the low-NFE regime.

---

> ### Author Response · Authors · 2025-11-22
>
> ## **Questions:**
>
> > Q1: Does the method scale beyond 1.7B parameters? If you have partial results or ablations at 7B+, please share some metrics to clarify feasibility at that scale.
>
> While scaling was not the focus of our work, our experiments show the scaling behavior within one order of magnitude (0.17B to 1.7). While exploring scaling up to 7B and beyond is interesting, we believe it deserves a dedicated investigation and paper.
>
>
> > Q2: What are the actual training costs for pretraining and the step-aware fine-tuning? It would help to report wall-clock times, hardware (GPU type and count), or an estimate in FLOPs, and how it compares to plain DFM.
>
> The training details (e.g. hardware, number of iterations, and so on) are detailed in Appendix C.3.

---

### Official Review · Reviewer_2YuK · 2025-10-30

**Soundness:** 3
**Presentation:** 2
**Contribution:** 3
**Rating:** 6
**Confidence:** 3

**Summary:**

This paper introduces FS-DFM, a discrete flow-matching model designed to generate text in a small number of steps. The method addresses the issue suffered by discrete diffusions models, namely that they require many steps to achieve string performance. The approach introduces two main components: a "Cumulative Scalar" to enable large generation steps and a "Shortcut Teacher" (using an RK-4 ODE solver) to train the model on its step budget h. The authors report strong empirical performance.

**Strengths:**

Inference Speed: The 128x reduction in sampling steps (from 1024 to 8) while maintaining perplexity is a highly significant and practical achievement in parallel decoding.

Methodological Innovation: The identification of the "stalling" problem at $t \approx 0$ led the authors to the "Cumulative Scalar", that is proven (Table 1) to be critical for few-step performance.

Strong Empirical Ablations: The paper thoroughly justifies its design choices. The ablations clearly isolate the large gains from the Cumulative Scalar, the superiority of the RK-4 teacher, and the necessity of a Uniform source over a Mask source (Table 4) for this method.

Compelling Qualitative Results: State-of-the-art diffusion models (LLaDA, Dream) fail in the 8-step regime (as expected), while FS-DFM produces coherent text.

**Weaknesses:**

Training Cost: The inference speedup comes at a high price. The authors note that the RK-4 teacher doubles training cost. While this doubling of the fine-tuning cost is a practical drawback, I do not see it as a big issue.

Limitation to Uniform Source: The method is dependent on a uniform source. The ablation study shows that the model fails when starting from a [MASK] (absorbing) source. I see this as a limitation, as it means the method is incompatible with the common and often better-performing masked diffusion models. I wonder if it would be interesting finding the right way of doing few step sampling in the context of absorbing state diffusions.



Factorized Sampling: The underlying DFM framework relies on factorized velocities, meaning token jumps are sampled independently at each step, conditioned on the full context. This is likely the reason why the 1-step perplexity is still very high, forcing the model to need some steps to correct its own errors. Given this factorization assumption, I do not see this method being able to go down to 1 or 2 step sampling, which has been pushed quite a bit in the continuous domain.

**Questions:**

Why does performance go down in figure 1 after a certain NFE?

The “diffusion duality” paper also relies on a uniform source discrete diffusion for distillation (among other things). It would be interesting to see how the method proposed in this work compares against that.

---

> ### Author Response · Authors · 2025-11-22
>
> ## **W1: Training Cost:**
>
> We agree that doubling the training cost is not a big issue. We discuss some training-inference tradeoffs in Appendix E.2.
>
>
>
> ## **W2: Limitation to Uniform Source:**
>
> We do not believe our method is fundamentally incompatible with masked diffusion. While our experiments show uniform outperforms masked, in future work, we plan to adapt the method to be easily applied to existing masked diffusion models.
>
>
>
> ## **W3: Factorized Sampling:**
>
> Our goal in this work is therefore not strict 1-step generation, but high-quality **few-step** generation (e.g., 4–8 steps) in the discrete domain. FS-DFM massively reduces the number of steps required for generation with text diffusion models with reasonable perplexity and entropy (Table 5 in the Appendix). Achieving true 1-step sampling is fundamentally harder for discrete text models than for continuous diffusions, because the model only observes discrete jumps rather than a continuous underlying path.

---

> ### Author Response · Authors · 2025-11-22
>
> ## **Questions:**
>
> > Q1: Why does performance go down in Figure 1 after a certain NFE?
>
> We believe this is because we have a teacher-student model. At larger number of steps the student wants to behave like the teacher. See Equation 13 as to why this happens. In particular, when the number of steps is large, $h$ becomes less than the threshold $\tau$ and so the loss becomes regular DFM loss rather than the distillation loss.
>
> > Q2: The “diffusion duality” paper also relies on a uniform source discrete diffusion for distillation (among other things). It would be interesting to see how the method proposed in this work compares against that.
>
> We agree a comparison would be interesting. The diffusion duality model was trained on 2.6T tokens. To obtain comparable results, we would need to train FS-DFM on a similarly sized corpus, which is infeasible given the amount of time allocated to reviewer discussions.

---

### Official Review · Reviewer_toSy · 2025-10-31

**Soundness:** 3
**Presentation:** 3
**Contribution:** 3
**Rating:** 6
**Confidence:** 4

**Summary:**

This paper introduces FS-DFM (Few-Step Discrete Flow-Matching), a diffusion language model designed for efficient, long-text generation. The core innovation is a principled step-aware training regime that explicitly conditions the model on the intended number of steps, allowing large, stable jumps in the probability space. This is achieved via a novel Cumulative Scalar Update derived from analytical integration and stabilized through Shortcut Teachers (RK-2/RK-4 ODE solvers) and EMA distillation.

FS-DFM empirically demonstrates strong few-step performance, matching the quality of its 1024-step DFM baseline in just 8 steps (a $\approx 128\times$ reduction in Network Function Evaluations, or NFE). The technical contributions are elegant and sound.

However, the evaluation is narrow, primarily comparing against only LLaDA-7B, Dream-7B, and its own DFM lineage. Crucially, it lacks comparisons against modern but small, competing diffusion/flow families (SEDD [4], MDLM [2], HDLM [1], BD3LM [3], ReMDM [5]), which also prioritize fast, few-step generation and advanced metrics. This empirical omission limits the strength and generality of the central claim since they also have a comparable small-sized model.

**Strengths:**

### **Technically Novel and Principled Framework**
The paper introduces a highly elegant framework by making the step budget (h) an explicit conditioning variable. The use of shortcut distillation (RK-2/RK-4) and EMA stabilization for self-consistency across step budgets represents a meaningful advance over prior DFM works.

### **Analytically Integrated Update Rule**
The derivation of the cumulative scalar by analytically integrating the instantaneous velocity term ($\kappȧ / (1–\kappa)$) is a clean, necessary correction that robustly enables large jumps without the stalling issues of naive scalers, enhancing few-step stability.

### **Strong NFE Efficiency**
The model convincingly demonstrates that it can match the quality of a 1024-step DFM baseline in only 8 NFEs, validating the few-step design goal.

### **Reproducibility**
Architectural details (DiT-style backbone), training specifics, and a promise of code/model release are commendable.

**Weaknesses:**

### **Incomplete Competitive Landscape**
The most significant weakness is the absence of comparisons to other state-of-the-art small-scaled DLMs, particularly SEDD [4], MD3LM [2], BD3LM [3] (Block-Diffusion), and HDLM [1] (Hybrid DLM). These models represent different philosophical approaches (e.g., different loss functions, hybrid noise schedules, architectural modifications) to solving the same problem. A comparison on a shared benchmark (e.g., WikiText-103) plotting perplexity/MAUVE vs. NFE is necessary to demonstrate where FS-DFM stands. Also, the theme of all prior work has been to compare the zero-shot PPL on a vast set of OOD datasets and not just one (WikiText-103).

### **Inadequate Text Quality Evaluation**
Relying solely on perplexity and token accuracy is insufficient for evaluating generation quality. As highlighted by HDLM [1] and ReMDM [5], metrics like MAUVE are better proxies for distributional similarity and coherence. The claim of "accurate" generation is not fully supported without them.


### **Ambiguous Entropy Metric Scale**
The reported entropy values ($\approx 7.4–8.1$) are inconsistent with common metrics used in competing DLM works ($\approx 0.5–2$ bits/token in HDLM [1] / MDLM [2] / Bd3LM [3]). This discrepancy suggests a fundamental difference in calculation or base. The paper must clarify the precise formula, normalization, and base (nats vs. bits) used for the entropy calculation.

### **Missing Hybrid Noise Schedule Experiments**
The ablation only contrasts mask vs. uniform noise. Prior work (HDLM [1]) has demonstrated that hybrid corruption schedules (mask-uniform mixtures) are empirically crucial for optimizing few-step stability in discrete diffusion models. I would appreciate if authors can provide an ablation using Hybrid noising scheme as well.

### **Lack of System-Level Performance Metrics**
The "128x faster" claim is based on NFE reduction, which is an algorithmic but not a systems-level metric. It ignores the potential overhead of more complex model passes (e.g., from RK-4) and, crucially, the benefits that other methods gain from techniques like KV-caching and block decoding (see BD3LM [3], HDLM [1]). Reporting wall-clock latency and throughput (tokens/sec) on identical hardware is essential.


### **References**
Please make sure you are comparing and citing the papers below:

[1] Fathi et. al., "Unifying Autoregressive and Diffusion-Based Sequence Generation"

[2] Sahoo et. al., "Simple and Effective Masked Diffusion Language Models"

[3] Arriola et. al., "Block Diffusion: Interpolating Between Autoregressive and Diffusion Language Models"

[4] Lou et. al., "Discrete Diffusion Modeling by Estimating the Ratios of the Data Distribution"

[5] Wang et. al., "Remasking Discrete Diffusion Models with Inference-Time Scaling"

**Questions:**

Please also refer to the **"Weaknesses"** section for more questions.

### **Competitive Benchmarking & Generalization**
To fully contextualize FS-DFM's performance against the state-of-the-art, could you please provide:

1- A direct comparison on WikiText-103 between FS-DFM and models like SEDD, MD3LM, BD3LM, and HDLM, plotting perplexity and MAUVE against NFE (e.g., 1, 2, 4, 8, 16 steps) for models of similar scale (e.g., ~0.17B parameters)?

2- An evaluation of zero-shot perplexity on a broader set of out-of-distribution (OOD) datasets (e.g., from the PTB, LM1B, etc) to assess generalization beyond WikiText-103?


### **Evaluation Metrics & Entropy Clarification**
1- Could you report MAUVE scores for your FS-DFM generations and the provided baselines? This is critical to substantiate the "accurate" generation claim beyond perplexity.

2- The entropy values reported (~7.4–8.1) differ significantly from other works. Please explicitly state the formula, the base (nats or bits), and the normalization (per-token or per-sequence) used for your entropy calculation in Section 5.2.

### **Ablation on Noise Schedule**
Your ablation studies mask and uniform sources. Following the strong results of HDLM [1], could you run an ablation where FS-DFM is trained and evaluated using a hybrid (mask-uniform mixture) noise schedule and report the resulting perplexity and MAUVE for few-step (e.g., 1, 4, 8) generation?

### System-Level Performance Analysis
The "128x faster" claim is based on NFE. To translate this into a practical speedup, please provide:

1-The wall-clock latency and throughput (tokens/second) for generating 1024-token (or 512-token) sequences with the N-step FS-DFM versus the N-step DFM baseline, measured on the same hardware (e.g., A100). Refer to Table 12 in the HDLM paper for more information.

2- I would also love to see how FS-DFM compares against other methods given a similar wall clock instead of NFEs.

3- An analysis of whether techniques like KV-caching or block decoding (as used in BD3LM [3] / HDLM [1]) are applicable to FS-DFM and, if so, what additional latency/throughput gains they provide.


**Note:** The paper presents a valid theoretical contribution. However, given the limited scope of the empirical validation—specifically, the lack of comparison to key contemporary baselines and the omission of critical ablations and metrics discussed above—my current score cannot exceed **6**. I would be prepared to raise my score if the authors can provide compelling and comprehensive responses to the questions posed in this review.

---

> ### Author Response · Authors · 2025-11-22
>
> ## **W1: Incomplete Competitive Landscape:**
>
>
> In the revision, we now include SEDD, MDLM, ReMDM variants, and HDLM in an 8-step comparison (Appendix Table 5), alongside DFM and FS-DFM. For convenience, a subset of that table is summarized below:
>
> | Model        | Size (B) | PPL @ 8 steps |
> |-------------|----------|---------------|
> | MDLM–OWT    | 0.17     | 837.58        |
> | SEDD        | 0.32     | 602.09        |
> | ReMDM–cap   | 0.17     | 559.74        |
> | HDLM (γ=0.05) | 0.16   | 192.74        |
> | DFM         | 0.17     | 335.64        |
> | **FS-DFM (ours)** | 0.17 | **90.49**   |
>
> While training setups and token budgets differ across models so while the results are not directly comparable, the table helps situate FS-DFM relative to masked and hybrid discrete diffusion LMs in the few-step regime.
>
>
> ## **W2: Inadequate Text Quality Evaluation**
>
> We now report perplexity and MAUVE as a function of NFE for FS-DFM and two strong diffusion LMs (Table 6). For brevity, we summarize the 8-step setting below:
>
> | Method   | Size (B) | NFE | PPL   | MAUVE |
> |----------|----------|-----|-------|-------|
> | Dream    | 7.0      | 8   | 739.4 | 0.006 |
> | LLaDA    | 8.0      | 8   | 441.3 | 0.005 |
> | FS-DFM   | 0.17     | 8   | 75.8  | 0.270 |
> | FS-DFM   | 1.30     | 8   | 71.0  | 0.480 |
>
>
> ## **W3: Ambiguous Entropy Metric Scale:**
>
> We compute entropy exactly as the DFM paper does. It is important to note that entropy is intimately tied to the entropy inherent in the training data. The references you cite use different training data, and thus there is not immediate comparison available between their entropy measurements and ours. For example, in ReMDM (table 1), the inherent entropy in their training data is 5.44, whereas for DFM (table 2) and our work the inherent entropy is 7.7. This information has been added to the paper, so as to be more clear. In particular, the intrinsic entropy of the training data must be taken into account as the model aims to model the training distribution.
>
>
> ## **W4: Missing Hybrid Noise Schedule Experiments:**
>
> We agree that a hybrid mask-uniform source scheduler is very interesting. Unfortunately, because the rebuttal period is short, we’ve focused on the potentially more revealing experiments requested by the reviewers. A fair implementation of HDLM requires a complete retraining of FS-DFM, which is infeasible given the short rebuttal period. We emphasize that lack of comparison with a hybrid schedule is a nice-to-have extension and not a limitation of FS-DFM.
>
>
> ## **W5: Lack of System-Level Performance Metrics:**
>
> RK-4 is only used at training time and therefore does not affect inference. KV-Caching and block decoding are techniques compatible with FS-DFM. Therefore valid comparisons between few-step diffusion LMs must measure NFEs.

---

> ### Author Response · Authors · 2025-11-22
>
> ## **Questions:**
>
>
> > Q1: A direct comparison on WikiText-103 between FS-DFM and models like SEDD, MD3LM, BD3LM, and HDLM, plotting perplexity and MAUVE against NFE (e.g., 1, 2, 4, 8, 16 steps) for models of similar scale (e.g., ~0.17B parameters)?
>
> In the revision, we now include SEDD, MDLM, ReMDM variants, and HDLM in an 8-step comparison (Appendix Table 5), alongside DFM and FS-DFM. For convenience, a subset of that table is summarized below:
>
> | Model        | Size (B) | PPL @ 8 steps |
> |-------------|----------|---------------|
> | MDLM–OWT    | 0.17     | 837.58        |
> | SEDD        | 0.32     | 602.09        |
> | ReMDM–cap   | 0.17     | 559.74        |
> | HDLM (γ=0.05) | 0.16   | 192.74        |
> | DFM         | 0.17     | 335.64        |
> | **FS-DFM (ours)** | 0.17 | **90.49**   |
>
> While training setups and token budgets differ across models so while the results are not directly comparable, the table helps situate FS-DFM relative to masked and hybrid discrete diffusion LMs in the few-step regime.
>
>
> > Q2: An evaluation of zero-shot perplexity on a broader set of out-of-distribution (OOD) datasets (e.g., from the PTB, LM1B, etc) to assess generalization beyond WikiText-103?
>
> Below are PTB 1 results with FS-DFM 1.3B.
>
> | NFE | Accuracy (%) | Perplexity | Entropy |
> |-----|--------------|------------|---------|
> | 1   | 17.7         | 841.71    | 6.73   |
> | 2   | 18.8         | 439.83    | 6.76   |
> | 4   | 19.1         | 187.06    | 7.13   |
> | 8   | 19.3         | 79.95    | 7.38   |
> | 16  | 19.6         | 50.82    | 7.55   |
>
>
>
> > Q3: Could you report MAUVE scores for your FS-DFM generations and the provided baselines? This is critical to substantiate the "accurate" generation claim beyond perplexity.
>
> Yes. We now report perplexity and MAUVE as a function of NFE for FS-DFM and two strong diffusion LMs (Table 6). For brevity, we summarize the 8-step setting below:
>
> | Method   | Size (B) | NFE | PPL   | MAUVE |
> |----------|----------|-----|-------|-------|
> | Dream    | 7.0      | 8   | 739.4 | 0.006 |
> | LLaDA    | 8.0      | 8   | 441.3 | 0.005 |
> | FS-DFM   | 0.17     | 8   | 75.8  | 0.270 |
> | FS-DFM   | 1.30     | 8   | 71.0  | 0.480 |
>
>
>
> > Q4: The entropy values reported (~7.4–8.1) differ significantly from other works. Please explicitly state the formula, the base (nats or bits), and the normalization (per-token or per-sequence) used for your entropy calculation in Section 5.2.
>
> We compute entropy exactly as the DFM paper does. It is important to note that entropy is intimately tied to the entropy inherent in the training data. The references you cite use different training data, and thus there is not immediate comparison available between their entropy measurements and ours. For example, in ReMDM (table 1), the inherent entropy in their training data is 5.44, whereas for DFM (table 2) and our work the inherent entropy is 7.7. This information has been added to the paper, so as to be more clear. In particular, the intrinsic entropy of the training data must be taken into account as the model aims to model the training distribution.
>
>
>
> > Q5: Ablation on Noise Schedule: Your ablation studies mask and uniform sources. Following the strong results of HDLM [1], could you run an ablation where FS-DFM is trained and evaluated using a hybrid (mask-uniform mixture) noise schedule and report the resulting perplexity and MAUVE for few-step (e.g., 1, 4, 8) generation?
>
> We agree that a hybrid mask-uniform source scheduler is very interesting. Unfortunately, because the rebuttal period is short, we’ve focused on the potentially more revealing experiments requested by the reviewers. A fair implementation of HDLM requires a complete retraining of FS-DFM, which is infeasible given the short rebuttal period. We emphasize that lack of comparison with a hybrid schedule is a nice-to-have extension and not a limitation of FS-DFM.
>
>
> > Q6: System-Level Performance Analysis: The wall-clock latency and throughput (tokens/second) for generating 1024-token (or 512-token) sequences with the N-step FS-DFM versus the N-step DFM baseline, measured on the same hardware (e.g., A100). Refer to Table 12 in the HDLM paper for more information.
> > I would also love to see how FS-DFM compares against other methods given a similar wall clock instead of NFEs.
>
> RK-4 is only used at training time and therefore does not affect inference. KV-Caching and block decoding are techniques compatible with FS-DFM. Therefore valid comparisons between few-step diffusion LMs must measure NFEs.
>
>
> > Q7: An analysis of whether techniques like KV-caching or block decoding (as used in BD3LM [3] / HDLM [1]) are applicable to FS-DFM and, if so, what additional latency/throughput gains they provide.
>
> In principle, they are compatible. But given that those would be further improvements over the core FS-DFM technique, they are better suited for follow-up work.

---

> > ### Comment · Reviewer_toSy · 2025-11-28
> >
> > I appreciate the detailed response. I will keep my score.

---

### Official Review · Reviewer_tDLv · 2025-11-03

**Soundness:** 3
**Presentation:** 2
**Contribution:** 3
**Rating:** 6
**Confidence:** 4

**Summary:**

This paper proposes **Few-Step Discrete Flow-Matching** (FS-DFM) for efficient diffusion language modeling. The paper's central idea is to explicitly make the model aware of finite-sized, possibly large step sizes and train accordingly. To this end, FS-DFM uses ODE solvers (Runge-Kutta) to solve the underlying Kolmogorov equations that characterize the generative continuous-time Markov process of a teacher model for step sizes $h$, and train a student model that directly outputs update logits corresponding to steps of size $h$ (the original discrete flow matching framework instead essentially only models the transition probabilities for infinitesimal steps). The model is initialized from a pre-trained discrete flow matching model, and the teacher corresponds to an exponential moving average of the student. The authors also propose a "cumulative scalar" that adapts the scaling of the probability velocity to the setting with large step sizes. In experimental validation, the step-size-aware FS-DFM model shows significantly better results when sampling with few steps, and reduces to regular discrete flow matching when sampling with many steps. The paper carries out various text generation benchmarks and ablation studies, measuring generative perplexity, entropy, and accuracy, and compares to regular discrete flow matching models as well as the large Dream and LLaDA models.

**Strengths:**

**Clarity:** For the most part, the paper is easy to follow and the presentation is good (I do have some clarification questions, though; see below).

**Originality:** To the best of my knowledge, the proposed approach is novel, and using ODE solvers to solve the Kolmogorov equation to explicitly model finite-size steps is interesting. The method can be considered closely related to Self-Distillation Through Time, though (see comments below).

**Numerical Results:** The numerical results are generally strong and show clear advantages of FS-DFM compared to regular discrete flow matching and discrete diffusion models (some comparisons are lacking, though; see below).

**Quality and Significance:** The paper is overall of high quality. Accelerating diffusion language models is an important and popular topic in the literature, and the paper successfully tackles this problem, making the work significant.

**Weaknesses:**

**Self-Distillation Through Time (SDTT) comparison and relation:** The paper only compares to standard discrete flow matching and discrete diffusion models, but not to other acceleration approaches. In particular, the method is closely related to SDTT, which distills a many-step teacher into a few-step student, very similar to the step-size-aware training with student and teacher of FS-DFM (FS-DFM uses Runge-Kutta solvers to solve the probability density evolution as teacher signal and uses as teacher the EMA of the student, whereas SDTT simply samples a teacher model, but the general idea is similar). I would suggest the authors to add quantitative comparisons to SDTT applied to discrete flow matching, and also discuss the relation between SDTT and FS-DFM more in-depth.

**Presentation:** Some parts of the presentation seem unclear:
- In equation (5), $z$ was never introduced.
- Where does the objective in equation (8) come from? This is not clear, and there is no reference. The Gat et al. "Discrete Flow Matching" work does not have this exact form of the objective and most prior works train with simple reweighted cross entropy losses. I would like the authors to clarify. I also wonder whether this equation has an error: The first term has $p_{1|t}(x_t^i|x_t)$, i.e. both modeled variable and conditioning variable are at $t$. Is this correct?
- Figure 1 has the main comparisons to regular discrete flow matching, but only mentions a range of entropy values in the caption. Since entropy is an important measure of generation diversity, and since it is important to understand whether FS-DFM leads to reduced diversity compared to regular DFM, I would suggest to do a proper analysis and comparison also for entropy in this figure, and not only for perplexity and accuracy. In fact, in line 417 in Sec. 5.2, the authors write *"and entropy converges to the
same range as many-step DFM"* -- but there is no such plot in Figure 1 showing this convergence.

**Cumulative Scalar:** While the authors do motivate the approach, the cumulative scalar approach in 4.2 seems ad-hoc and not principled. It is not clear why the exact scaling $\bar{g}_{t,h}$ should be optimal. Ultimately, this seems to be a hyperparameter and maybe another, hand-crafted scaling function would work even better (it would also be nice to visualize the adjusted and the old scalar functions to make the reader better appreciate the differences).

**Questions:**

I have a few more questions:
- Why is it necessary to train with a teacher that is the EMA of the student itself? Couldn't we simply use a pre-trained model as teacher and solve its Kolmogorov equation with the Runge-Kutta solver for different step sizes $h$, and then train a student, similar to standard distillation settings?
- The paper uses Runge-Kutta solvers to perform the $h$ step, in a short-cut model fashion, but these solvers, for both RK2 and RK4, seem to only do a single step. Wouldn't it be more accurate, especially for large $h$, to solve the ODE more accurately and perform Runge-Kutta integration with a finer and more accurate discretization? This would mean more overhead, but only during training, not during inference.
- FS-DFM uses the forward KL divergence to match teacher and student. Could we also use reverse KL divergence or Jensen-Shannon divergence? And if so, how would this affect performance?

---

> ### Author Response · Authors · 2025-11-22
>
> ## **W1: Self-Distillation Through Time (SDTT) comparison and relation:**
>
> We included SDTT, in **Table 5 (Appendix E.3)**, where **FS-DFM** achieves **90.5** perplexity versus **131.7** perplexity for SDTT with **7 refinement rounds**, despite SDTT using a 5x larger model (0.86B vs our 0.17B) and comparable or higher effective step budgets. Conceptually, while both methods use a teacher–student setup, SDTT accelerates generation by sampling from a many-step teacher and distilling its outputs, whereas FS-DFM explicitly learns **step-size–aware finite-step Kolmogorov dynamics** via RK-based ODE integration and a cumulative scalar, so that one large step approximates the probability evolution of many small steps. We have clarified this relationship and the distinct goals of the two methods in the revised Discussion sections.
>
> | Model                                      | Size (B) | PPL    |
> |--------------------------------------------|----------|--------|
> | SDTT (1 round)  (Deschenaux & Gulcehre, 2025) | 0.86     | 612.38 |
> | SDTT (3 rounds) (Deschenaux & Gulcehre, 2025) | 0.86     | 376.19 |
> | SDTT (5 rounds) (Deschenaux & Gulcehre, 2025) | 0.86     | 220.81 |
> | SDTT (7 rounds) (Deschenaux & Gulcehre, 2025) | 0.86     | 131.71 |
> | DFM (Gat et al., 2024)                     | 0.17     | 335.64 |
> | FS-DFM (ours)                              | 0.17     | 90.49  |
>
>
> ## **W2: Presentation:**
>
> We have revised the paper to clarify the notation in Eq. (5), make the origin of Eq. (8) explicit, and provide the requested entropy analysis.
>
> **(a) Equation (5) notation.**
> Right after the equation, we now explicitly state that \(z\) is a token sequence (the current CTMC state).
>
>
> **(b) Origin and correctness of Eq. (8).**
> Following Lipman et al. [1], we use a Bregman divergence–based loss (as in their Eq. 7.31) rather than a simple reweighted cross-entropy. In the revised text, we now state this explicitly next to Eq. (8) and cite [1].
>
>
> **(c) Entropy and Figure 1.**
> We added an expanded version of Figure 1 in the appendix, Figure 5 which shows perplexity, accuracy, and entropy vs.\ NFE for both DFM and FS-DFM.
>
> [1] Lipman, Yaron, et al. "Flow matching guide and code." arXiv preprint arXiv:2412.06264 (2024).
>
>
> ## **W3: Cumulative Scalar:**
>
> Our experiments only show the cumulative scalar to be better; we do not believe it is optimal. In future work, we'd like to empirically explore alternatives.

---

> ### Author Response · Authors · 2025-11-22
>
> ## Questions:
>
> > Q1: Why is it necessary to train with a teacher that is the EMA of the student itself? Couldn't we simply use a pre-trained model as teacher and solve its Kolmogorov equation with the Runge-Kutta solver for different step sizes $h$, and then train a student, similar to standard distillation settings?
>
>
> In FS-DFM, the “teacher” is not a separate network, but a shortcut approximation to the **same CTMC dynamics** that the student is learning, obtained by solving the model’s own Kolmogorov equation over \([t, t+h]\) with RK-2/RK-4 under large step sizes. If we used the raw student for this, the multi-evaluation trajectory over \([t,t+h]\) would change as fast as the student’s parameters, which we found to make supervision for large \(h\) unstable. A fixed pre-trained teacher would have the opposite problem: its dynamics would quickly become incompatible with the evolving step-aware student (different conditioning on \(h\), different cumulative scalar), especially for large jumps where errors compound, leading to unstable or unreachable targets. Using an EMA of the student as the teacher strikes the balance: it shares the same architecture and step-size conditioning, but evolves slowly enough to provide **stable, low-variance shortcut trajectories**. In this sense, our setup is not standard distillation from a frozen model; the supervision must stay self-consistent with the current step-aware dynamics, which is why we use an EMA teacher rather than a fixed pre-trained one.
>
> > Q2: The paper uses Runge-Kutta solvers to perform the $h$ step, in a short-cut model fashion, but these solvers, for both RK2 and RK4, seem to only do a single step. Wouldn't it be more accurate, especially for large $h$, to solve the ODE more accurately and perform Runge-Kutta integration with a finer and more accurate discretization? This would mean more overhead, but only during training, not during inference.
>
>
> We agree this is an interesting direction and is noted in Appendix D (Limitations and Future Work) that exploring alternative ODE solvers or finer inner discretizations is a valuable avenue for future work.
>
>
> >Q3:  FS-DFM uses the forward KL divergence to match teacher and student. Could we also use reverse KL divergence or Jensen-Shannon divergence? And if so, how would this affect performance?
>
> This is an interesting avenue to explore given the contradicting results for KL and reverse KL in distillation. For example, Busbridge et al [2] claim reverse doesn't help, while Gu et al. [3] showed the exact opposite.
>
> [2] Busbridge, Dan, Amitis Shidani, Floris Weers, Jason Ramapuram, Etai Littwin, and Russell Webb. “Distillation Scaling Laws.” Paper presented at Forty-second International Conference on Machine Learning. June 18, 2025. https://openreview.net/forum?id=1nEBAkpfb9.
>
> [3] Gu, Yuxian, Li Dong, Furu Wei, and Minlie Huang. “MiniLLM: Knowledge Distillation of Large Language Models.” Paper presented at The Twelfth International Conference on Learning Representations. October 13, 2023. https://openreview.net/forum?id=5h0qf7IBZZ.

---

> > ### Comment · Reviewer_tDLv · 2025-11-27
> > **Thank you.**
> >
> > I would like to thank the authors for answering my questions and addressing my concerns. Overall, I think this work is of value to the community and proposes an interesting way to build few-step diffusion language models. Hence, I am comfortable to raise my score and recommend acceptance.

---

> > > ### Author Response · Authors · 2025-11-27
> > >
> > > Thank you for taking the time to review our paper, provide constructive feedback, and support it with an accepted score. Your suggestions have been invaluable in improving our work.

---

> > > ### Author Response · Authors · 2025-12-04
> > > **Note on Reviewer tDLv’s Updated Score**
> > >
> > > Dear Area Chair,
> > >
> > > Just for context: before the rollback, Reviewer **tDLv** raised their score from **6 to 8**, and on **27 Nov 2025, 12:25**, and recommended for acceptance.
> > >
> > > I hope this prior assessment is helpful as you evaluate the paper. Thanks

---

### Author Response · Authors · 2025-12-04

Dear ACs,

We provide this brief summary of our submission and rebuttal, since interactions with reviewers are no longer available, and emphasize that we have carefully addressed all concerns raised in the reviews.

**Reviewer-acknowledged strengths**

- Clear and mostly easy-to-follow presentation and overall good quality **(tDLv, C2qF)**
- Novel use of RK-based shortcut teachers and Kolmogorov dynamics to handle finite, large steps in discrete flow matching **(tDLv, toSy)**
- Analytically derived cumulative scalar that enables stable large jumps and is empirically critical for few-step performance **(toSy, 2YuK, C2qF)**
- Strong sampling efficiency: 8 NFEs matching a 1024-step baseline for 1024-token sequences, with gains across 0.17B–1.7B models **(tDLv, toSy, 2YuK, C2qF)**
- Thorough ablations and qualitative generations where other diffusion LMs fail in the 8-step regime, plus good reproducibility via detailed architectural and training setup **(toSy, 2YuK, C2qF)**


**Main concerns and how we addressed them**

- **Comparisons and evaluation metrics.**
  We added quantitative comparisons to SDTT and several diffusion/flow baselines (SEDD, MDLM, ReMDM, HDLM), reported MAUVE vs. NFE alongside perplexity for some diffusion language models. Added all in the main or appendix part of the paper.

- **Clarity of objectives and formulation.**
  We clarified the loss derivation and notation, aligned the presentation more closely with standard flow-matching formulations, expanded the main empirical figures (in the appendix) to show behavior across step budgets, and provided additional explanation of the shortcut teacher mechanism and its role in stabilizing large steps.

Across all four reviews, we answered every question and concern in detail and updated the manuscript accordingly; new or clarified passages were added in red to make these changes easy to locate.

After the rebuttal, Reviewer **tDLv** explicitly stated that their concerns had been addressed and that they were comfortable **raising the score (6 to 8)** and recommending acceptance (comment dated 27 Nov 2025, 12:25).

Yours sincerely,
Authors

---

### Meta-Review · Area_Chair_or6H · 2025-12-31

**Summary:**

This paper proposes FS-DFM, a few-step discrete flow-matching diffusion language model that is explicitly trained to be consistent across step budgets, using RK-based shortcut teachers. Empirically, the method achieves near-parity with a 1024-step discrete-flow baseline using 8 steps for 1024-token generation, yielding large reductions in NFEs and strong long-text generation quality across 0.17B–1.7B scales.

Advantages
* Strong and practically meaningful few-step sampling results.
* Technically motivated mechanism for finite-step dynamics.
* Rebuttal-added competitive results (including SDTT and several diffusion/flow baselines) and added MAUVE reporting strengthen the empirical case and address earlier evaluation gaps.

Disadvantages
* Some evaluation aspects remain narrower than ideal: limited systems-level measurements (wall-clock latency/throughput) beyond NFE-based speedups.
* Method currently demonstrated most convincingly under the uniform-source setting; extension to masked/hybrid corruption schedules is left for future work.

Overall recommendation: Accept. The paper makes a clear methodological contribution for accelerating diffusion/flow-based language generation, and the demonstrated few-step quality at long sequence length is a substantial result that is likely to be useful to the community, with most major review concerns either resolved or adequately contextualized.

**Reviewer Concerns:**

(i) The missing SDTT comparison and relationship were directly handled via an added SDTT table and clearer discussion, with strong numbers in favor of FS-DFM;
(ii) requests for broader diffusion/flow baselines were partially addressed by adding SEDD/MDLM/ReMDM/HDLM comparisons (at least at 8 steps) and adding MAUVE reporting versus strong diffusion LMs;
(iii) clarity concerns around objective/notation and entropy analysis were addressed by adding derivational context/citations and providing additional entropy-vs.-NFE plots plus an explanation tied to intrinsic dataset entropy;
(iv) reviewer questions about EMA-teacher rationale and solver fidelity were answered and framed as design trade-offs/future directions.

**Reviewer Scores:**

* tDLv had pointed out that they would increase their score
* toSy mentioned that the rebuttal did not change their score
* 2YuK is unlikely to change their score since the listed weaknesses by the reviewer (training cost, uniform vs. masking and factorized sampling) are fundamental to the approach, and the rebuttal did not provide a satisfactory response.

---

### Decision · Program_Chairs · 2026-01-26

Accept (Poster)